# Lipid Peroxidation in Muscle Foods: Impact on Quality, Safety and Human Health

**DOI:** 10.3390/foods13050797

**Published:** 2024-03-04

**Authors:** Stefan G. Dragoev

**Affiliations:** Department of Meat and Fish Technology, Technological Faculty, University of Food Technologies, 26 Maritza Blvd., 4002 Plovdiv, Bulgaria; s_dragoev@uft-plovdiv.bg

**Keywords:** meat, poultry, fish, shelf life, lipid oxidation, mechanisms, health risk, hazards

## Abstract

The issue of lipid changes in muscle foods under the action of atmospheric oxygen has captured the attention of researchers for over a century. Lipid oxidative processes initiate during the slaughtering of animals and persist throughout subsequent technological processing and storage of the finished product. The oxidation of lipids in muscle foods is a phenomenon extensively deliberated in the scientific community, acknowledged as one of the pivotal factors affecting their quality, safety, and human health. This review delves into the nature of lipid oxidation in muscle foods, highlighting mechanisms of free radical initiation and the propagation of oxidative processes. Special attention is given to the natural antioxidant protective system and dietary factors influencing the stability of muscle lipids. The review traces mechanisms inhibiting oxidative processes, exploring how changes in lipid oxidative substrates, prooxidant activity, and the antioxidant protective system play a role. A critical review of the oxidative stability and safety of meat products is provided. The impact of oxidative processes on the quality of muscle foods, including flavour, aroma, taste, colour, and texture, is scrutinised. Additionally, the review monitors the effect of oxidised muscle foods on human health, particularly in relation to the autooxidation of cholesterol. Associations with coronary cardiovascular disease, brain stroke, and carcinogenesis linked to oxidative stress, and various infections are discussed. Further studies are also needed to formulate appropriate technological solutions to reduce the risk of chemical hazards caused by the initiation and development of lipid peroxidation processes in muscle foods.

## 1. Introduction

For over a century, researchers in the field of food chemistry have grappled with one of the most significant issues: understanding the patterns in changes that the lipid fraction of foods undergoes when exposed to atmospheric oxygen. This process, inherently spontaneous, goes by various names such as autoxidation, denaturation, or lipid peroxidation. Lipid peroxidation is a process established in the organism of living animals, which develops and propagates during the subsequent slaughtering, the carcass meat cutting and deboning, and its next technological processing. Subsequently, these processes extend through technological processing, storage, and even during culinary preparation [1].

The reactivity of lipids with atmospheric oxygen is determined by the degree of unsaturation of the polyunsaturated fatty acids (PUFA) present in their molecular structure. Oxidative processes can be initiated or inhibited by the presence of non-polar or polar lipids, incl. and of phospholipids, sterols, pigments, fat-soluble vitamins, metals of variable valence, and other compounds found in muscle tissue.

From a theoretical perspective, lipid peroxidation manifests as a complex chain radical process. Furthermore, it holds practical significance in relation to the formation of stable, biologically active compounds that are incorporated into muscle foods [2].

It is widely recognised that the quality and safety of meat and fish undergo alterations due to enzymatic or non-enzymatic catalysed hydrolysis, oxidation processes, and the presence of putrefactive microorganisms. The transformations observed in muscle lipids from meat and fish, induced by oxygen radical species during extraction, processing, and storage, are linked to the deterioration of sensory properties, degradation of biologically active compounds, diminished nutritional value, and the accumulation of primary and secondary derivative products that can be harmful to human health [3].

## 2. The Essence of lipid Peroxidation in Muscle Foods

### 2.1. Phases of Lipid Peroxidation

Lipid peroxidation is distinguished by a chain radical mechanism, a process encompassing distinct phases of initiation, propagation, and termination in the chain reaction [4].

The initiation process commences with the abstraction of a proton, leading to the formation of an alkyl radical (L·) as depicted in Reaction (1). This radical then engages with oxygen, giving rise to a peroxyl radical (LOO·) as outlined in Reaction (2). Subsequently, the peroxyl radical extracts hydrogen from unsaturated fatty acids, resulting in the generation of a hydroperoxide (LOOH), which stands as a significant primary product in the autooxidation reaction, as articulated in Reaction (3) [5].
(1)$$Initiation\;LH\;\overset{\mathrm{initiator}}{\to }L\cdot$$
(2)$$Prolongation\;L\cdot +\;O_{2}\;\to \;LOO\cdot$$
(3)LOO+LH → LOOH+L·
(4)Prolongation LOOH → LO·+ HO·
(5)$$2LOOH\;\to \;LOO\cdot +\;LO\cdot {+\;H}_{2}O$$
(6)$$\begin{array}{l} \;Termination\;LO\cdot +\;LO\cdot \to \\ \;\text{non-radical}\;polymers\;\\ \;LOO\cdot \;+LOO\cdot \to \end{array}$$
where LH—unsaturated fatty acid; HO·—hydroxyl radical; L·—alkyl radical; LO·—alkoxyl radical; LOO·—peroxyl radical, and LOOH—lipid hydroperoxide. 

### 2.2. The Cyclical Nature of Lipid Peroxidation

A spin barrier exists, hindering direct access to atmospheric oxygen in a triplet primary condition for the molecules of polyunsaturated fatty acids. In a singlet basic condition, the formation of an alkyl radical (L·) is not hindered by this spin barrier [4].

Numerous potential initiators and propagators of lipid peroxidation in muscle foods are recognised, including the hydroxyl radical (HO·), perferryl and ferryl radicals, Fe^2+^-O_2_-connected radicals, and porphyrin cation radicals (P-Fe^4+^ = O). Additionally, enzyme systems such as lipoxygenases, cyclooxygenases dependent on nicotinamide adenine dinucleotide phosphate (NADPH), adenosine diphosphate–Fe^3+^ (ADF–Fe^3+^) and O_2_ enzymes play a role in this process [5]. The complete understanding of which form of iron—free or connected, haem or non-haem, oxidised or reduced—possesses the capacity to oxidise polyunsaturated fatty acids in meat remains elusive [3]. A pivotal aspect of lipid peroxidation revolves around identifying the primary catalysts that initiate and propagate the process (Figure 1) [1], along with recognising its cyclic nature (Figure 2) [3].

## 3. Mechanisms of Free Radical Initiation in the Lipid Fraction of Muscle Foods

Several mechanisms for initiating free radicals from lipids in muscle tissue are known [6]. Oxygen species and activated catalysts emerge as the primary initiators of the oxidation of polyunsaturated fatty acids across all three described reaction pathways [7] (Figure 3).

### 3.1. Non-Enzymatic Initiation of Lipid Peroxidation in Meat Products

The precise mechanisms by which non-enzymatic initiated lipid peroxidation occurs in muscle foods are still the subject of scientific debate [8]. Drawing upon the literature references pertaining to other biological systems, a number of probable mechanisms can be traced [9]. The production of reactive oxygen species such as the superoxide anion radical (O_2_^−^·) and hydroxyl radical (HO·) is crucial for initiating and propagating the process in meat and fish products. The superoxyl anion radical (O_2_^−^·) can be generated through several different mechanisms:(1)By oxidases, such as cytochrome oxidase, catalysing the transfer of electrons from cytochrome to oxygen;(2)Through auto-oxidation of oxy-myoglobin and oxyhaemoglobin: (both containing Fe^2+^ in the oxidised state) with the formation of superoxyl anion radical (O_2_^−^·) and metmyoglobin or methaemoglobin (both containing Fe^3+^ in the oxidised state);(3)Via free iron ions capable of participating in transfer reactions with molecular O_2_, leading to the generation of superoxyl anion radical (O_2_^−^·) [10].

The superoxide anion radical (O_2_^−^·) is not reactive enough to abstract a proton from lipids, although it can lead to the formation of H_2_O_2_. The peroxide anion radical can be produced through various enzymatic reactions in mitochondria, microsomes, and sarcoplasm, where its substrates are present [11]. It is an important precursor for the generation of hydroxyl radicals, which can initiate lipid peroxidation. Simple iron complexes (Fe^2+^.ADP and their salts) can react with H_2_O_2_ to form a hydroxyl radical (HO·), which initiates lipid peroxidation by extracting hydrogen from another lipid radical. In this way, the chain reaction spreads. It is accepted that until iron is released from heme proteins, they cannot react with H_2_O_2_ or the superoxide anion radical (O_2_^−^·) to form hydroxyl radical (HO·). To initiate the lipid peroxidation process, iron has to be liberated from these proteins [11].

### 3.2. Enzymatic Initiation of Lipid Peroxidation in Meat Products

Initiation of lipid peroxidation occurs through microsomal and mitochondrial enzymatic factors. The initiation of lipid peroxidation involves enzymatic factors from both microsomes and mitochondria. In red meat, lipid peroxidation can be enzyme-initiated by various enzyme factors [2,3,12]. The microsomal fraction of poultry and fish skeletal muscles contains enzyme systems that catalyse lipid peroxidation in the presence of co-factors [13,14,15]. A similar catalyst system has been identified in both pork and beef microsomal lipids [13,16]. Enzyme-initiated lipid peroxidation in microsomes in skeletal muscles depends on either NAD or NADPH. In the former case, it requires ADP and Fe^2+^, while in the latter, it involves ADP and Fe^3+^. The initiation of lipid peroxidation commences in the membranes because membrane phospholipids predominantly contain unsaturated fatty acids [17,18,19]. Iron binds to the surface of membrane proteins and phospholipids, catalysing the initiation of lipid peroxidation. There is evidence suggesting that the Fenton reaction may not occur with the participation of iron in a non-proton environment [20,21], such as the inner layers of the membranes or the surface of triacylglycerol droplets [22]. The activity of microsomal enzyme lipid peroxidation is more pronounced in beef muscles separated from the *post-rigour* carcass compared to *pre-rigour*, particularly in oxidative muscle types (m. *Trapezius*) compared to glycolytic ones (m. *Longissimus dorsi*) [23]. Enzyme systems initiating lipid peroxidation are isolated from both microsomes and mitochondria [17,18,19].

Xanthine oxidase/dehydrogenase (EC 1.17.3.2; Xanthine: O_2_ oxidoreductase) is an iron–molybdenum flavoprotein (FAD) containing (2Fe-2S) centres with a molecular mass of 283,000. It belongs to the group of aerobic dehydrogenases, functioning as an oxidoreductase. This group of two-component oxidoreductases, including xanthine oxidase, contains flavin–adenine–mononucleotide (FAM) or flavin–adenine–dinucleotide (FAD). Enzymes with these non-protein components can reduce oxygen to hydrogen peroxide, classifying them as aerobic dehydrogenases. Xanthine oxidase is composed of two identical subunits and possesses a complex prosthetic group, consisting of two molybdenum atoms, two malls of FAD, eight iron atoms, and eight acid atoms in proportions of 1:1:4:4. The primary substrates of xanthine oxidase are hypoxanthine and xanthine, which undergo oxidation with the participation of oxygen and water. Molecular oxygen, colourants, and nitrates can also serve as electronic acceptors in similar reactions. The enzyme is abundantly present in the liver, kidneys, mucous cells of the small intestine (jejunum), heart, lungs, and intestines in mammals. In tissues, xanthine oxidase is interconvertible with xanthine dehydrogenase D-form, with this form transforming into the oxygen O-form. Reactions catalysed by xanthine oxidase result in the formation of the superoxyl anion radical (O_2_^−^·) and hydrogen peroxide (H_2_O_2_) [24].

Hematin and haem proteins, including myoglobin, haemoglobin, cytochrome with oxidase, and peroxidases, have the capability to initiate lipid peroxidation processes [25].

Cytochrome c oxidase (EC 1.9.3.1.; cytochrome C: O_2_ oxidoreductase) is a transmembrane protein composed of six to thirteen protein subunits, two heme groups (a and a3), and two copper atoms. The molecular mass of the enzyme is approximately 200 kDa. This enzyme belongs to the group of oxidases and serves as an oxidoreductase, which is the most prevalent oxidase in biological systems. Unlike enzymes that utilise FMN or FAD as cofactors, cytochrome c oxidase is distinctive in that it transfers electrons directly to oxygen. It functions exclusively in aerobic environments, requiring the presence of oxygen to oxidise substrates. Therefore, it is categorised as an aerobic enzyme [26]. The oxidation of cytochrome c in the presence of the enzyme is represented by the reaction (7): (7)$$4\;cytochrome\;c\;({Fe}^{2+})+4H^{+}+\;O_{2}\to 4\;cytochrome\;c\;({\mathrm{Fe}}^{2+})+2H_{2}O$$

Indeed, the process is a more intricate chain reaction, where electrons extracted from cytochrome are sequentially transferred through cytochromes *a* and *a3* to oxygen, involving the participation of ferric and cupric ions.

Cytochrome P450 reductase (EC 1.6.2.4; cytochrome P450: O_2_ oxidoreductase) and activated ADP-Fe^3+^ play a crucial role in NADPH-dependent lipid peroxidation occurring in muscle microsomes, underscoring the significance of cytochrome *P450* reductase [27]. During microsomal transfer reactions, the reduction in oxygen results in the formation of reactive oxygen species [26]. Electron transfer from NADP cytochrome *P450* reductase to iron-binding compounds can be facilitated directly by EDTA-Fe^3+^ and DTPA-Fe^3+^ or indirectly through peroxide [27]. Hydroxyl radicals (HO·) are implicated in NADP cytochrome *P450* reductase ADP-Fe^3+^-dependent microsomal lipid peroxidation [27]. Hydroxyl radicals (HO·) are generated through the reduction in hydrogen peroxide (H_2_O_2_) in an iron-catalysed Haber–Weiss reaction [28].

Microsomal lipid peroxidation, dependent on NADP cytochrome *P450* reductase and xanthine oxidase/ADP/Fe^3+^, is initiated by the reduction in ADP-attached iron, facilitated by NADP cytochrome *P450* reductase or superoxyl anion radical (O_2_^−^·), and a reaction termed the “pumping” of electrons. This process leads to the formation of ferric ions [29].

The activation of oxygen by cytochrome *P450* involves the formation of various oxygen complexes. In the absence of electrons, the oxygen complex releases a peroxyl radical (LOO·), which isomerises to hydrogen peroxide (H_2_O_2_). Another electron withdrawal from the oxygen complex of cytochrome *P450* generates an active oxygen complex [27]. The formation of water from the active oxygen complex of cytochrome *P450* is attributed to the inclusion of a radical of thiol cationic Fe^4+^ = O chemical compounds [30].

Catalase (EC 1.11.1.6; H_2_O_2_: H_2_O_2_ oxidoreductase) belongs to the group of oxidases, specifically peroxidases, functioning as an oxidoreductase. Comprising four protein subunits, each containing a protohemin group, catalase is thermolabile and loses its activity at 35 °C. Consequently, the action of catalase alters the quality of chilled or frozen meat/fish exclusively.

In muscle tissue and meat, the activated catalase enzyme plays a crucial role in breaking down hydrogen peroxide (H_2_O_2_) into water (H_2_O) and oxygen (O_2_). This enzymatic activity serves as a natural barrier, effectively impeding the initiation and propagation of lipid peroxidation [31].

Peroxidase (EC 1.11.1.7; donor: H_2_O_2_ oxidoreductase) denotes a broad category of enzymes that oxidise substrates with the involvement of hydrogen peroxide (H_2_O_2_) as a hydrogen acceptor. This enzyme group can be categorised into iron-containing peroxidases, including Fe^3+^ protoporphyrin and Fe^2+^ porphyrin, and flavin-dependent peroxidases, which feature an FAD prosthetic group. Peroxidases exhibit various activities, including peroxidase activity, oxidising activity, catalase activity, and hydroxylase activity. While peroxidases are sensitive to heat, they are resilient to low temperatures and maintain their activity even at −18 °C [32].

The interaction between peroxidases and hydrogen peroxide (H_2_O_2_) activates the prosthetic ferric protohemin enzyme to a higher redox state, enabling it to attack various compounds, including lipids [33]. The active intermediate in the peroxidase-catalysed reaction comprises one oxygen atom, added by peroxide, coordinative linked to the haem iron. This complex is typically oxidised two units higher than the residual ferry enzyme and is best described as a porphyrin cation radical, an iron–oxygen complex (Fe^4+^ = O), and P–Fe^4+^ = O [34].

The role of glutathione peroxidase (EC 1.11.1.9.; glutathione: H_2_O_2_ oxidoreductase) in initiating lipid peroxidation in muscle tissue has been well established for a considerable period [35]. Peroxidase-catalysed lipid peroxidation constitutes a crucial step in the biosynthesis of the hormone thyroxine and plays a significant role in defining biological mechanisms [36].

Activated protohemin proteins play a role in the mechanism of lipid peroxidation, which is initiated by protohemin compounds through the haemolytic cleavage of preformed fatty acid hydroperoxides to free radicals. Protohemin proteins catalyse the initial development of lipid peroxidation but are not the direct initiators of this process [37]. The interaction of hydrogen peroxide with metmyoglobin or methaemoglobin rapidly generates an active chemical species that triggers membrane lipid peroxidation [38]. The activated protohemin protein, functioning as an initiator of lipid peroxidation, is a porphyrin cation radical (P-Fe^4+^ = O) [39]. Notably, activated metmyoglobin and methaemoglobin are considered the actual initiators of lipid peroxidation. Despite sharing the same prosthetic group as peroxidases and catalase, their interaction with hydrogen peroxide (H_2_O_2_) occurs at a lower rate. This interaction results in the formation of a free radical [40]. Monahan et al. [41] proposed that the haem protein radical of the metmyoglobin–hydrogen peroxide system exhibits reactivity similar to that of peroxidase.

Two other enzymes, lipoxygenases and cyclooxygenases, also play a role in initiating lipid peroxidation in meat and fish [42]. These enzymes have the ability to directly introduce an oxygen atom into the carbon chain of lipid molecules, thereby stimulating lipid peroxidation processes [43]. Lipoxygenases and cyclooxygenases belong to the group of oxidoreductases known as oxygenases. This enzyme group catalyses oxidation–reduction reactions involving the direct attachment of oxygen to the substrate, leading to the formation of hydroxyl or carboxyl groups. Specifically, those enzymes that add a molecule of oxygen are referred to as dioxygenases and fall under subclass 1.13.

Lipoxygenase (EC 1.13.11.12.; linoleate: O_2_ 13S-oxidoreductase) catalyses the formation of lipid peroxides as the final products of the reaction [44] and thus initiates lipid peroxidation of meat [45]. These enzymes utilise long-chain fatty acids with a cis, cis-(1-4)-pentadiene unit (–CH=CH–CH_2_–CH=CH–) [46] as substrates, such as linolenic, arachidonic, eicosapentaenoic, and docosahexaenoic acids (at meat and fish), and oxidise them to conjugated hydroxydiene derivatives—specifically hydroperoxides [47]. Multiple lipoxygenase isomers exist, differing primarily in the positional specificity of hydrogen withdrawal and the addition of oxygen to the substrate fatty acid. The product of 5-lipoxygenase, 5-hydroperoxyeicosatetraenoic acid, serves as the immediate precursor of leukotrienes. Lipoxygenase-like enzymes have been found in chicken muscles [47]. Oxidised products of arachidonic acid are produced in the presence of 15-lipoxygenase [47]. According to [45], raw beef loin and chicken thigh had higher lipoxygenase-like activity compared to chicken breast and pork loin. Animal lipoxygenases catalyse a reaction producing lipid hydroperoxides [46]. During their decomposition, they form secondary oxidation products responsible for the *warmed-over flavour* of meat [46]. Lipoxygenase can directly oxidise PUFA from membrane phospholipids and generate lipid hydroperoxides [48]. Ferry myoglobin generated from the interaction of metmyoglobin and H_2_O_2_ and the hydroxyl radical through the Fenton reaction directly abstracts a hydrogen atom from PUFA and generates lipid hydroperoxide, similar to lipoxygenases [48]. It was hypothesised that ferry myoglobin is primarily responsible for lipoxygenase activities [48]. Lipoxygenase activity is responsible for the biosynthesis of eicosanoids from arachidonic acid in the cell membranes of beef or pork loin or chicken breast and legs [48]. After cooking, lipoxygenase-like activities in meat are significantly reduced [48]. The relatively high lipoxygenase-like activity of beef loin was found to be due to the higher concentration of myoglobin and the lower concentration of reducing compounds because ascorbic acid is able to reduce ferry myoglobin [45]. Lipoxygenase activity and enzyme-catalysed lipid oxidation in dry-cured hams are dependent on the temperature, NaCl, and pH [49].

Cyclooxygenase (EC 1.13.11.31.; arachidonate: O_2_ 12-oxidoreductase) can be effectively activated by substrate analogues such as long-chain fatty acids [50]. The catalytic mechanism of cyclooxygenases is similar to that of lipoxygenases. Many unsaturated fatty acids, including linolenic, act as potent inhibitors. The substrate specificity depends on the levels of surrounding hydroperoxides, referred to as “peroxide tone”. The eicosapentaenoic acid is a strong inhibitor at low levels of peroxides (<10^−8^ M), whereas at high levels of peroxides (>10^6^ M), it becomes a substrate for the enzyme reaction [51,52].

Enzymes can also play a role in the formation of reactive oxygen species, such as hydrogen peroxide (H_2_O_2_) or the superoxyl anion radical (O_2_-·). Particularly discussed in fish is the enzyme *lipoxygenase (EC 1.13.11.12.)* [53]. It initiates lipid peroxidation processes in the sarcoplasm or microsomes and exhibits both sarcoplasmic and membrane-bound activity [54]. Lipoxygenase specifically catalyses oxidation at the sixth carbon atom of both n-6 and n-3 fatty acids. For instance, linoleate 13 (S)-lipoxygenase from sardine skin [55] and 15-lipoxygenase from trout gills [56] have been isolated. Gill and skin tissues of fish also contain 12-lipoxygenase, which is active on arachidonic acid and the n-3 isomers of eicosapentaenoic and docosahexaenoic acids [57,58,59,60]. Linoleate 13 (S)-lipoxygenase has a pH optimum at pH 7.0 and initiates lipid peroxidation in fish [58]. Mackerel lipoxygenase has been found to oxidise linoleic and docosahexaenoic acids more efficiently than eicosapentaenoic and linolenic acids [61].

To initiate the formation of free radical chains under the action of lipoxygenase, an activation reaction is required to start the enzyme [60]. The development of a rancid taste in American emerald shiner (*Notropis atherinoides*) [53] and rainbow trout (*Salmo irideus*) [59] is indirectly associated with the enzymatic action of lipoxygenases located in the skin and gills of fish. Josephson et al. [62] confirmed the presence of the same n-9 lipoxygenase in the gills of saltwater and freshwater fish. It reacts with n-9 and n-12 unsaturated fatty acids and forms 12 (S)-hydroperoxyl fatty acid isomers [56]. Purification of the n-9 lipoxygenase revealed another n-6 active lipoxygenase. Under the action of n-9 and n-6 lipoxygenases in fish, 12-hydroperoxide of arachidonic acid and 14-hydroperoxide of docosahexaenoic acid are formed [57]. Heat treatment inactivates lipoxygenase and enhances autoxidative reactions in fish. When lake herring are heated to 40 °C, about 80% of the lipoxygenase activity is lost, and after 20 s holding at 60℃, about 90% of the enzyme activity is lost [63]. Fish lipoxygenases retain high activity at negative temperatures [55,64].

Another enzyme present in fish is *cyclooxygenase (EC 1.13.11.31.)*. It forms prostaglandins by stereo, specifically introducing oxygen into arachidonic acid [42]. The reduction in oxygen and the formation of active oxygen products occur during electron transfer reactions in microsomes. The microsomal enzyme system of fish requires the presence of coenzyme NAD/NADP and iron. Two electrons are transferred from NAD/NADP through NAD/NADP cytochrome *P450* reductase and the superoxyl anion radical to iron-binding compounds. A complex of Fe^3+^ donates electrons, and the ferrous ions thus formed activate lipid peroxidation. The reaction is enhanced in the presence of ADP and ATP [65]. It is likely that the function of ADP is to incorporate ferric ions into a complex, thereby keeping them soluble so that they can be reduced by the NAD/NADP system [66]. The ferrous complex can be activated by oxygen and form a ferryl radical (ADP-Fe^4+^ = O·), which initiates lipid peroxidation. The fish cyclooxygenase system exhibits relatively high activity at relatively low temperatures compared to enzyme systems from warm-blooded livestock and birds [67].

Lipid peroxidation in both light and dark herring muscle can be initiated by an *NAD-dependent enzyme system* in its microsomal fraction. The enzyme from dark muscles is more active. Both enzyme systems have a pH optimum of 6.7, with ferrous ions stimulating the enzyme reactions more effectively than ferric ions [55]. NADP-dependent lipid peroxidation has also been found in the microsomes of American channel catfish (*Ictalurus punctatus*). It develops at −10 °C, and a temperature lower than −20 °C has been recommended to extend the shelf life of channel catfish fillets. Any thawing results in the reactivation of fish peroxidase systems [68]. The enzyme-catalysed microsomal lipid peroxidation reaction in winter halibut (*Pseudopleuronectes americanus*) muscle is limited to the formation of lipid hydroperoxides. Copper ions catalyse this type of lipid peroxidation, and the reaction shows no selectivity for NAD or NADP [69]. Enzymatic lipid peroxidation in both light and dark muscle microsomes from herring (*Clupea harengus*) is dependent on ATP or ADP, NAD, and iron [70]. In the soluble fraction of muscle juice of mackerel (*Scomber scombrus*), 8% of the iron and 7–38% of the copper are associated with fractions with a molecular mass less than 5 kDa, and the rest are in a fraction with a molecular weight above 5 kDa. It has been suggested that both fractions could initiate lipid peroxidation in mackerel muscle [71]. Two types of factors suppress the development of lipid peroxidation in American rainbow trout (*Salmo gardenrii*) sarcoplasm. One type is heat-labile and non-dialysable, and the other is heat-labile and dialysable. Both factors are glutathione-independent and protect polyunsaturated fatty acids in fish microsomes [72,73].

## 4. Free Radical Chain Mechanisms for Propagation of Oxidative Processes in Muscle Foods

The discussion on the mechanisms for the formation of malondialdehyde (MDA) in diene and the triene system is not new and dates back to the middle of the twentieth century [74,75]. Lipid hydroperoxides are highly unstable, and when they break down, they form various secondary products. These secondary products are responsible for the appearance of a rancid taste in meat and fishery products. The degradation of hydroperoxides proceeds through a free radical mechanism [76], creating opportunities for interactions between alkyl radicals (L·) and other free radicals or molecules. The formation of various secondary products, such as carboxyl compounds, alcohols, acids, and aldehydes, including MDA, occurs [74,75,77].

MDA is a three-carbon aldehyde that is obtained through the autooxidation of PUFA. Linolenic acid produces much more MDA compared to the autooxidation of linoleic acid. No MDA is formed from the autooxidation of oleic acid. Dahle et al. [78] propose that linoleic acid does not produce MDA in the early stages of autooxidation. They suggest that PUFA containing a triene system undergoes autooxidation, resulting in peroxide radicals that saturate the β and γ positions to the carbon associated with a peroxide group (Figure 4).

The radical compound (**1**) can cyclise and form a five-atomic cyclic peroxide radical compound (**2**), which is capable of extracting hydrogen from the alkyl group and forming compound (**3**) or undergoing peroxidation by releasing hydrogen and forming compound (**4**). Compounds (**3**) and (**4**) may form MDA in an acidic medium or through heat treatment (Dahle et al. [78]). Thus, the proposed mechanism does not include the release of the initial hydrogen atom and subsequent formation of peroxyl radicals in the allylic positions at the ends of the diene system, as indicated in Figure 5.

The release of hydrogen from the allylic positions leads to the formation of compound (**5**), a peroxyl radical with non-violence at the β and γ positions to the carbon atom associated with the peroxyl radical (LOO·). This radical (**5**) could cyclise and form compound (**6**), and then undergo further interaction by forming compounds (**7**) and (**8**). In the presence of heat or acid treatment, these compounds would then form MDA. Thus, following the mechanism proposed by Dahle et al. [78], the autooxidation of dienes gives considerable amounts of MDA.

Pryor et al. [79] offer a modification of the mechanism proposed by Dahle et al. [78] for a more adequate explanation of the formation of MDA by PUFA (Figure 6). Compound (**2**), whose formation is shown in Figure 4, undergoes a cyclical closure (ring formation) and produces cyclic endo peroxyl radical compound (**10**), which subsequently undergoes peroxidation and the release of alkyl hydrogen, resulting in compounds (**11**) and (**12**). MDA forms when any one of these compounds is exposed to heat or acid. In the diene system, compound (**6**) forms a double cyclic endo peroxyl radical (**9**) (Figure 5).

This radical is about 10 kcal less stable than the allyl radical (**10**) (Figure 6). It is expected that radical (**9**) would form more easily than radical (**10**), making the formation of MDA easier in a diene system [80]. The mechanism proposed by Pryor et al. [79] explains more adequately why the oxidised products of PUFA produce much more MDA than fatty acids with one or two double bonds. This mechanism is similar to the one proposed for the synthesis of prostaglandins, in which MDA is a secondary product [5,81].

## 5. Kinetics of Oxidation Processes

In living organisms, oxidation reactions are rigorously regulated. In instances of antioxidant deficiency or subsequent death and subsequent technological processing, the standard control mechanisms for lipid oxidation become disrupted. This phenomenon is linked to the presence of active oxygen species that instigate lipid peroxidation [82].

### 5.1. Electron Structure of Oxygen

The electron structure of oxygen comprises two non-binding electrons of π anti-binding energy level in a triplet state 3Σg. Consequently, the reaction of oxygen with molecules in a singlet multiple basic state 1Σg, such as PUFA, is precluded by a spin (Table 1). This limitation does not apply to reactions involving single electrons, hydrogen atoms, or other atoms and molecules containing unrelated electronic pairs, such as metals with variable valence or free radicals [1,13]. The reduction in oxygen with four electrons to water results in a highly positive potential (E^0^ = +0.82 V vs. NHE at pH 7.0). The conversion of this potential through one-electron reduction processes yields various products, including superoxyl anion radical (O_2_^−^·), hydroperoxyl radicals (HO_2_^−^·), hydrogen peroxide (H_2_O_2_), and hydroxyl radicals (HO·) [13,24,83]. The theory of molecular orbitals elucidates the presence of triplet oxygen in a basic state with a double bond. Peroxyl radicals (LOO·) exhibit similar bonds but with only one unrelated electron of anti-binding π level. The addition of hydrogen with an affiliated proton or hydrogen atom to oxygen results in the formation of hydroperoxyl radicals (HO_2_^−^·) or hydrogen peroxide (H_2_O_2_). They manifest π binding, and by hybridisation sp^3^—sp^3^, two additional σ-bonds are formed [13,83,84]. In this scenario, oxygen undergoes p^3^ hybridisation and possesses two pairs of unpaired valence electrons.

The reduction of a single electron hydrogen peroxide (H_2_O_2_) results in the generation of species with almost the same number of electrons of anti-binding and binding levels. These species prove to be unstable and undergo decay, forming hydroxyl anions (HO^−^) and hydroxyl radicals (HO·). The molecule’s disintegration prompts a new hybridisation of the sp^3^ orbital, and both entities adopt a tetraeder structure [13,83,84]. The redox potentials of oxygen and its reduced species (E°) in an aquatic environment are as follows:(8)$$E{}^{\circ}\;pH\;{4.0\;O}_{2}\overset{-0.05}{\to }{HO}_{2}\cdot \overset{+1.44}{\to }H_{2}O_{2}\overset{+0.71}{\to }H_{2}O\;+\cdot OH\;\overset{+2.81}{\to }H_{2}O$$
(9)$$\begin{array}{l} E{}^{\circ}\;pH\;{7.0\;O}_{2}\overset{-0.33}{\to }O_{2}\cdot \overset{+0.89}{\to }H_{2}O_{2}\overset{+0.30}{\to }H_{2}O+OH\cdot \overset{+2.81}{\to }H_{2}O\\ {\;2H}^{+}{\;H}^{+} \end{array}$$

Highly positive redox potentials, such as that exhibited by HO·, signify the molecule’s robust prooxidant nature, capable of oxidising nearly all known biomolecules. The oxygen molecule’s lowest energy electron configuration, comprising two electrons in π anti-binding orbitals (excluding the triplet ^3^Σg basic state), can exist in two excited singlet states, particularly in a ^1^Δg state, which boasts a significantly longer lifespan than the ^3^Σg state [14]. 

Within cell organelles and muscle tissue, xanthine oxidase, aldehyde oxidase, and NADPH oxidase can generate superoxyl anion radical (O_2_^−^·) in the range of 0.1–1.0 μM [13,82,83,84]. Additionally, these radicals can be produced through the auto-oxidation of oxyhaemoglobin and oxymyoglobin [85]. In the oxidation of Fe^2+^, superoxyl anion radical (O_2_^−^·) can also be formed [86,87].
foods-13-00797-t001_Table 1Table 1Energy levels of singlet oxygen (Cotton and Willkins [88]).**Type****π*a****π*b****Energy, kJ**^1^Σg
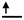

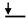
155^1^Δg
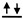

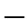
92^3^Σg
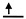

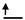
0—basic state*Note:* π*—electrons in antibonding level. 


### 5.2. Superoxyl Anion Radical

The sources of superoxyl anion radical (O_2_^−^·) in meat include electron transfer systems in membranes, autoxidation of oxymyoglobin to metmyoglobin, activation of some leukocytes, and the presence of ascorbic acid and other reducing components facilitating the release of “free” iron [84].

From a thermodynamic standpoint, the superoxyl anion radical (O_2_^−^·) functions as a potent oxidant, yet it cannot directly accept an electron due to the instability of the resulting oxygen [4,7,83,84]. This kinetic barrier significantly restricts its ability to directly oxidise organic molecules. While the superoxyl anion radical (O_2_^−^·) cannot directly oxidise lipids, it can indirectly deactivate catalase and peroxidases, interact with compounds of these enzymes [11], or oxidise α-tocopherol [89]. Consequently, it impedes the protective action of these compounds against the propagation of lipid peroxidation. The conjugate acid of the superoxyl anion radical (O_2_^−^·), the peroxyl radical (LOO·), with a pKa in the water of 4.8, serves as a much stronger oxidant (E° = +1.44 V) than the superoxyl anion radical (O_2_^−^·). The peroxyl radical (LOO·) initiates the chain oxidation of linoleic and arachidonic acids [90]. The approximate rate of oxidation of linoleic acid by the peroxyl radical (LOO·) is 300 M^−1^.s^−1^ at pH = 4.7. In biological systems, nearly 0.3% of the formed oxygen exists in the protonated form. Due to the electrical double layer with predominant negative charges on the membrane surface, their pH is 3.0 units lower than that of the surrounding sarcoplasmic environment [91]. The change in pH depends not only on the surface charge, which varies in different membranes but also on the ionic strength. Therefore, the conversion of the superoxyl anion radical (O_2_^−^·) to the peroxyl radical (LOO·) can occur in proximity to the membrane, and the loss of charge facilitates the radical’s penetration into the lipid zone. Here, the peroxyl radical (LOO·) can initiate oxidative reactions and disrupt membrane functions [42]. In muscle tissue, with the *post mortem* decrease in pH from initial values of 6.80–7.00 to final values of 5.4–6.0 during *post mortem* numbness (*rigor mortis*), the proportion of peroxyl radical (LOO·) can reach 10–20% relative to the amount of superoxyl anion radical (O_2_^−^·) [92].

### 5.3. Hydrogen Peroxide

Hydrogen peroxide (H_2_O_2_) is typically found as a metabolite in small concentrations within aerobic cells. The concentration of hydrogen peroxide (H_2_O_2_) is regulated to be around 10^−11^–10^−7^ M with the assistance of hydroperoxide isomerase and glutathione peroxidase [93]. Mitochondria, microsomes, peroxisomes, and sarcoplasmic enzymes are recognised as efficient generators of hydrogen peroxide (H_2_O_2_) when their substrates are available [4,5]. Hydrogen peroxide (H_2_O_2_) can be directly formed by various enzymes, such as L-α-hydroxy acid oxidase, L-gluconolactone oxidase, aldehyde oxidase, D-amino oxidase, glucose oxidase, etc. [94]. Additionally, hydrogen peroxide (H_2_O_2_) can be obtained non-enzymatically through the autoxidation of phospholipid flavins, thiols, and other reducing compounds, by the spontaneous isomerisation of superoxyl anion radical (O_2_^−^·), or with the aid of hydroperoxide isomerase. The enzymatic isomerisation of the superoxyl anion radical (O_2_^−^·) at physiological pH is 10^6^ times faster than the spontaneous process [91]. Although hydrogen peroxide (H_2_O_2_) is not a strong oxidant (E° is 0.30 at pH = 7.0) and is not reported to react directly with PUFA [8,11,95], through one-electron reduction, it transforms into a powerful oxidising compound—the hydroxyl radical (HO·), with an E° is 2.18 at pH = 7.0. The hydroxyl radical (HO·) is capable of oxidising lipids [11,13,95].

Fenton and Jackson [96] initially reported that ferrous ions (Fe^2+^) strongly promote the oxidation of polyhydric alcohols in the presence of hydrogen peroxide (H_2_O_2_). Haber and Weiss [97] suggested that the hydroxyl radical (HO·) is the actual oxidant in the Fenton reaction. Subsequently, “Fenton reagents” were demonstrated to generate hydroxyl radicals (HO·) [98]. The Fenton reaction is illustrated by Equation (10):(10)$${Fe}^{2+}+{\;H}_{2}O_{2}\to {Fe}^{3+}+{HO}^{-}+HO\cdot$$

A reaction between the superoxyl anion radical (O_2_^−^·) and hydrogen peroxide (H_2_O_2_) does occur, but it possesses a very low rate constant and is too slow to be considered an efficient or significant source of the hydroxyl radical (HO·) [99]. Consequently, the notion that the hydroxyl radical (HO·) is formed in the so-called Haber–Weiss reaction is discarded. 

### 5.4. Hydroxyl Radical

The highly active hydroxyl radical (HO·) is readily generated in oxygen-generating systems in the presence of iron or copper ions through an iron-catalysed Haber–Weiss reaction mechanism [2,3,100]:(11)$${Me}^{n+}+O_{2}\to {Me}^{(n-1)+}+O_{2}$$
(12)$${Me}^{(n-1)+}+H_{2}O_{2}\to {Me}^{n+}+{HO}^{-}+\mathrm{HO}\cdot$$
(13)$${Pure\;O}_{2}^{-}+H_{2}O_{2}\overset{Metal}{\underset{Catalyst}{\to }}O_{2}{+HO}^{-}+HO\cdot$$

The reduction in ferric ions by ascorbate occurs more rapidly than its reaction with the superoxyl anion radical (O_2_^−^·) [101]. In biological systems where superoxyl anion radical (O_2_^−^·) or hydrogen peroxide (H_2_O_2_) is formed with iron-containing proteins like lactoferrin [102], transferrin [103], ferritin [104], the formation of a hydroxyl radical (HO·) becomes possible [105]. However, there is a contrasting perspective suggesting that transferrin and lactoferrin cannot stimulate the formation of hydroxyl radical (HO·) from superoxyl anion radical (O_2_^−^·) or hydrogen peroxide (H_2_O_2_) [106,107]. Transferrin can catalyse the formation of a hydroxyl radical (HO·) after the release of oxygen by neutrophil NADP oxidase in the presence of hydrogen peroxide (H_2_O_2_).

### 5.5. Iron Ions and Iron Complexes

Iron ions and iron complexes play a crucial role as initiators of oxidative processes. They can catalyse lipid peroxidation in the presence of thiol groups and ascorbic acid [108]. Hydroxyl radicals are implicated in the initiation of lipid peroxidation in these reactions [109]. The oxidation of arachidonic acid to conjugated dienes by xanthine oxidase depends on both superoxyl anion radicals and hydrogen peroxide, with hydroxyl radicals being responsible for the initiation of lipid peroxidation [110]. Lipid peroxidation can be initiated by ferrous ions (Fe^2+^) without the addition of reducing compounds [1,13,111], by reactions (14)–(17):(14)$${Fe}^{2+}+O_{2}\to {Fe}^{3+}+O_{2}^{-}\cdot$$
(15)$${HO}_{2}^{-}+{HO}_{2}\cdot \to H_{2}O_{2}+O_{2}^{-}\cdot$$
(16)$${Fe}^{2+}+H_{2}O_{2}\to {Fe}^{3+}+{HO}^{-}+HO\cdot$$
(17)$$LH+HO\cdot \to L\cdot +\;{HO}^{-}$$

Redox compounds serve as the primary driving force in acquiring higher concentrations of ferrous ions, which, in turn, generate hydroxyl radicals (HO·) and initiate instigate lipid peroxidation. This process is recognised as iron redox cycling catalysis of lipid peroxidation [112].

### 5.6. Singlet Oxygen

Singlet oxygen can be generated through microwave discharge, chemical reactions, and photochemical reactions [75]. Light-sensitive molecules (S) initially create an electronically excited species (^1^S). This singlet excited species has an extremely short lifetime, lasting only 10^−11^ s. It dissipates its energy by emitting a photon through interaction with the solvent, forming a new excited state—a triplet electronically excited state (^3^S). The triplet sensitiser has a sufficiently extended lifetime to interact with other molecules and initiate photochemical reactions [113,114]. The triplet sensitiser (^3^S) then directly reacts with triplet oxygen (^3^O_2_), which is in its usual ground state, thereby forming singlet oxygen through reactions (18) and (19)
(18)S0→S1→S3
(19)S3+O23→S0+O12

There are two excited states of singlet oxygen: a higher energy state ′Σg and a lower energy state ′Δg. The ′Δg state possesses 92 kJ.mol^−1^ of added energy and is the sole singlet oxygen species that actively engages in reactions in solution. Singlet oxygen ′Δg is electrophilic and swiftly reacts with molecules featuring highly substituted double bonds or electron-rich functional groups, such as unsaturated fatty acids, aromatic or sulphur amino acids, and purines [113,114]. The initiation of the oxidation of unsaturated fatty acids by singlet oxygen follows a mechanism distinct from that of free radicals. Singlet oxygen can directly react with double bonds through addition, a process known as the “ol” reaction [24,75].

Nawar [115] reported that from linolenate, which has two 1,4-pentadiene structures, two pentadienyl radicals can be produced after hydrogen abstraction from the two active methylene groups at carbon atoms 11 and 14. As a consequence of oxygen attack at the terminal carbon of each of these radicals, a mixture of 9-, 12-, 13-, and 16-hydroperoxides is formed. Nawar [115] is of the opinion that geometric isomers are known for each of these hydroperoxides with the conjugated diene system in cis-, trans- or in trans-, trans- configuration. On the other hand, their isolated double bond is always in the cis configuration.

Compared to the 12- and 13-hydroperoxides, the 9- and 16-isomers are formed in greater amounts. Cyclisation is the most likely mechanism for this specificity of the reaction. This may be due to the fact that oxygen has a greater affinity to react with carbon atoms 9 and 16. On the other hand, 12- and 13-hydroperoxides decompose more rapidly and tend to form six-membered hydroperoxides via 1,4–cyclisation. Another possibility is that the 12- and 13-hydroperoxides decompose by 1, 3-cyclisation similar to prostaglandin endoperoxides [115].

### 5.7. Haemoproteins

Haemoproteins accelerate the formation of peroxyl radicals (LOO·) through the disproportionation of the singlet oxygen and electronically excited states of the carbonyl group [11,116] as depicted by reactions (20) and (21):(20)LOO·+ LOO·→LOH+LO+O21
(21)$$LOO\cdot +\;LOO\cdot \to LOH+LO\cdot +\;O_{2}$$

Widely distributed in biological tissues, metals with variable valence, such as iron and copper, with their labile d-electron system, serve as catalysts for redox reactions [117,118]. These metals exhibit a broad range of accessible oxidation states, enabling electron transfer [119,120]. The redox potential for such transfer is variable and depends on the stereochemistry of the ligands. Stable paramagnetic states, resulting from the presence of unpaired electrons, are common for variable valence metals, facilitating their reaction with radical substrates [121]. Consequently, they can alleviate spin constraints between oxygen and PUFA and promote lipid peroxidation [121]. Transition metals can initiate the process of lipid peroxidation in various mechanisms [95,122]:(1)By forming a radical of PUFA, for instance, through a reaction catalysed by lipoxygenase, involving the transfer of one electron or the evolution of hydrogen;(2)By generating a superoxyl anion radical (O_2_^−^·) and indirectly interacting with triplet oxygen, thus forming a more reactive oxygen species;(3)By oxidising phospholipid flavin cofactors, they activate oxygen by adopting a semiquinone radical state, indirectly participating in the generation of an oxygen species;(4)By interacting with oxygen (using cytochrome *P450*) or with peroxides (via peroxidase, catalase, myoglobin, haemoglobin, or cyclooxygenase).

By transferring an oxene from the oxygen or peroxide to the metal ion, the formal valence of the metal is increased from 3+ to 5+. The formation of oxocations, such as Fe^5+^ = O_2_^−^, is characteristic of the chemical behaviour of iron, manganese, molybdenum, and cobalt. Gilbert [123] distinguished three types of reactions:(1)Direct initiation by Fenton-type reactions;(2)Indirect initiation, via hypervalent iron complexes;(3)Indirect initiation and propagation, via iron-catalysed degradation of hydroperoxides to peroxyl radicals (LOO·) and abstraction of hydrogen from unsaturated fatty acids.

Direct initiation can occur in three mechanisms:(1)Fe^3+^ abstracts H^+^ from unsaturated fatty acids and forms an alkyl radical (L·);(2)Fe^2+^ forms metal–oxygen transfer complexes, generating reactive oxygen species in non-polar solvents—an unlikely mechanism for meat products;(3)Metal autooxidation, which produces reactive oxygen species through the Haber–Weiss reaction supplemented by a Fenton reaction [124,125].

Indirect initiation is facilitated by incorporating iron hypervalent complexes, often referred to as active forms of hem proteins and porphyrin compounds. These complexes are formed with the assistance of enzymes such as peroxidase, cytochrome *P450*, catalase, and others. Haemoproteins, such as myoglobin and haemoglobin, directly catalyse lipid peroxidation [126,127].

The third type of reaction involves mechanisms of both indirect initiation and the development of lipid peroxidation. Both haem and non-haem iron in ferrous and ferric states can catalyse the degradation of the presented hydroperoxides to alkoxyl and peroxyl radicals (LO·, LOO·). These radicals abstract H+ from unsaturated fatty acids. The breakdown of hydroperoxides leads to a chain reaction, as these radicals abstract hydrogen at a higher rate than alkyl radical (L·) [128].

Approximately two-thirds of the iron in the body is incorporated into haemoglobin, with an additional 10% found in myoglobin. Small amounts of iron, in the form of a prosthetic component, are distributed among various iron-containing enzymes and the transport protein transferrin. The remaining iron is housed within intracellular proteins, specifically ferritin and hemosiderin. A minor reservoir of iron, not bound to proteins and moving between transferrin, cell sarcoplasm, mitochondria, and ferritin, constitutes the so-called “free” iron [129]. “Free” iron has been identified in tissues at micromolar concentrations [106].

Direct Fe^3+^ electronic attack on most organic molecules in aqueous solutions is only feasible for compounds with very low redox potential, for example, phospholipid flavines and thiols, with redox potentials ranging from −0.5 to 0 V, or some enols such as ascorbic acid or polyphenols, with redox potentials from 0 to +0.5 V. Biomolecules like hydrocarbons, unsaturated fatty acids, or aromatic compounds, with redox potentials greater than +1.00 V, can only be oxidised by metals with redox potentials exceeding +1.00 V [130]. Many metal-containing enzymes exist in a low redox state, where their redox potentials hover around zero [131]. Higher redox potentials are achieved in metal-containing enzymes acting as intermediates during complex oxygen binding or other oxygen species [132].

The 3d orbitals of Fe^3+^ are in a high spin state. Upon forming a ligand, they adopt a low spin arrangement at the ^t^2_g_ level. This configuration allows the eg orbital to remain unoccupied for the ligands’ electrons, forming an iron coordination bond. This arrangement reduces the oxidative potential of the ferric ligand, such as Fe^3+^-EDTA, compared to the “free” iron ion during ascorbic acid oxidation [133]. It elevates the redox potential of the ferrous ligand Fe^2+^-EDTA in comparison with the ferrous ion during the reduction of hydrogen peroxide (H_2_O_2_) to hydroxyl radicals (HO·) [134].

An exceedingly crucial form of iron is the *haem group*, where four of the octahedral orbitals are ligand-connected to the porphyrin ring. Haem-containing cytochromes, such as those found in proteins, exemplify this property. In these proteins, the iron is situated in one plane with four ligands derived from the porphyrin ring. Only two ligands, which are perpendicular to this plane, influence the chemical relations and binding of iron. Haem-containing proteins exhibit significant variations in their redox potential. This diversity enables cytochromes to function as highly specific carriers of reducing equivalents at various redox potentials. Numerous iron-containing enzyme systems and pigments play crucial roles in cellular biochemical processes. Notably, lipoxygenase, cyclooxygenase, cytochrome *P450*, peroxidases, myoglobin, and haemoglobin can directly or indirectly initiate membrane lipid peroxidation [134].

*Light* has been demonstrated to accelerate the oxidation of lipids in pork and turkey meat, a process that can be suppressed by singlet oxygen ^1^O_2_ inhibitors [2,43,91].

## 6. Oxidative Stability of Muscle Lipids

### 6.1. Background

It has been demonstrated [91] that the initiation of auto-oxidative processes of lipid peroxidation in cell membranes is brought about by the attack of hydrogen peroxide (H_2_O_2_), free hydroxyl radical (HO·) or superoxyl anion radical (O_2_^−^·) on phospholipids and PUFA. The oxidative stability of skeletal muscles is primarily dependent on three factors [135]: the type of oxidised substrates, the presence and activity of catalysts for lipid peroxidation, and the presence and activity of antioxidants and prooxidants. Compensatory mechanisms play a crucial role in maintaining the balance among these factors, which are responsible for the control of oxidation in skeletal muscles [136,137] (Figure 7). However, during processing, this balance is disrupted [1,2,5,15].

Once lipid peroxidation progresses to the propagation phase, an unpleasant rancid smell and taste, known as *warmed-over flavour* (WOF) [16,53], become evident [20]. Numerous internal and external factors within animal tissues have been identified as influencing the rate of *post mortem* lipid peroxidation [135,138].

Internal factors are the biological species [139], the anatomical location and type of muscles [140], the fatty acid composition and triacylglycerol structure of muscles and/or fat [138,141], the presence of natural pro- and antioxidant systems [17,28,138], the initial degree of lipolysis and lipid oxidation in the meat raw materials, the content of free fatty acid residues at the commencement of technological treatment [84,142], and so forth.

External factors include the system of animal feeding, the addition of pro- and antioxidant compositions to the forages [138,143], the presence of oxygen during meat processing [1,95], the type of technological processing, packaging, and storage [144,145], the temperature [84,138,146,147], light or other types of irradiations [84,144,148,149,150], the addition of salting materials, antioxidants, and more, various types of nutritional supplements [13,15,20,28,149], the packaging method or the type of packages [13,84,144,151], etc.

### 6.2. Type of Oxidised Substrates

One approach to enhancing the oxidative stability of meat and fish involves modifying oxidised substrates [135]. Two such substrates subject to alteration in meat are the fatty acid profile of muscle lipids [138,143,150] and the oxygen content of cells [1,95]. Initially, lipid peroxidation in meat occurs at the level of cell membranes [13,84,95,144]. Igene and Pearson [152] propose importing fat with greater oxidative stability through the animal diet. The reduction in the proportion of unsaturated fats in the feed results in the oxidative stabilisation of endogenous lipids. However, this approach is undesirable in terms of nutritional value and the texture of muscle foods.

The fatty acid profile of endogenous lipids plays a crucial role in determining the speed of lipid peroxidation in muscle foods [56,143,151,153]. Saturated fatty acids are practically inert in this process [154]. The activation energy required to rupture the C-H bond of monounsaturated fatty acids (MUFA) is significant. Conversely, the more unsaturated the meat fat, the more unstable and susceptible to the initiation and propagation of lipid peroxidation [8,11,13,16,84]. More effective control of lipid peroxidation could be achieved by replacing endogenous with exogenous lipids. The fatty acid composition of pigs, birds, and fish can be altered by changing the sources of lipids in their fodder [138,143,151]. Reducing the percentage of saturated fatty acids results in meat with greater oxidative stability, but this is undesirable in terms of biological and nutritional value and the tenderness and juiciness of the meat. Lipid peroxidation can also be influenced by limiting oxygen access during technological processing [1,95]. It is recommended that meat or fish be processed under vacuum or in a modified atmosphere [155,156,157]. Vacuum packaging is an effective method to limit the oxidation of meat and fish products [155,156]. To prevent the development of lipid peroxidation and the occurrence of *warmed-over flavour* (WOF), it is imperative to vacuum pack cooked meat and fishery products as soon as possible after their rapid cooling at 0–4 °C [148,158,159].

### 6.3. Prooxidant Factors

Several technological operations can disrupt the oxidative balance of muscle tissue. These operations include:-Deboning, tendon removal, shaping, chopping, grinding, mincing, and cutting, are processes in which lipid peroxidation catalysts and substrates are mixed [138]. Consequently, oxygen enters the anaerobic muscle tissue;-Salting increases the catalytic activity of iron and reduces antioxidant enzyme activity [13,15,28,150];-Heat treatments such as surface hot drying, roasting, hot smoking, steaming, boiling, grilling, baking, and frying [1,13,15,138] destroy the cellular organisation of muscles, leading to protein and enzyme denaturation. This, in turn, affects the antioxidant enzymatic activity, which is partially or completely lost, releasing iron connected with proteins.

To limit lipid peroxidation in muscle foods, technologies should be applied to maintain or improve the oxidative balance existing in the raw whole muscles. This can be achieved by reducing the pro-oxidant effect of technological equipment, altering the concentration of oxidised substances, and employing antioxidants [135].

Prooxidant action of high temperatures. The prooxidant action of high temperatures is notable. Heat aids in releasing oxygen from oxymyoglobin [13,145,160], forming active hydrogen peroxide (H_2_O_2_) at 60 °C [106] and at 4 °C [40]. Active hydrogen peroxide degrades haem structures [42,84] and releases “free” iron [13,95]. With the increase in temperature, more oxygen is liberated into the muscle tissue [39]. During heat treatment, hydroxyl radicals (HO·) are generated [7,13,161], and hydroperoxides break down into free radicals [40,84,162].

Baking chickens with skin in a convection oven at 177 °C to the centre temperature of 78 °C, followed by 4 days of storage at 4 °C, significantly increases the amount of free malondialdehyde (MDA), accumulating faster in the skin than in the muscles [163]. Similarly, baking chicken breasts and legs in a microwave oven at 162.7 °C and in a conventional oven at 176.6 °C, to a centre temperature of 65.6 °C, and subsequent 2 days of storage at 4 °C, also leads to an increase in free MDA [164]. Notably, free MDA accumulates faster in legs roasted for a longer time at a lower temperature.

Reheated H_2_O_2_-activated myoglobin from a horse’s cardiac muscle was found to exhibit the most pronounced pro-oxidant activity at 74 °C, but as the temperature increased to 100 °C it decreased [165].

Prooxidant action at low temperatures. Prooxidant action at low temperatures is noteworthy. Lipid peroxidation decelerates upon freezing [166,167]. Lipid hydroperoxides, being soluble in fat, exhibit increased stability at low temperatures [106,168]. To extend the shelf life of meat and fish, it is advisable to maintain the temperature as low as possible [169].

Prooxidant activity in mechanically separated red meat. Prooxidant activity has been observed in mechanically separated red meat. Initially, Kunsman et al. [20] identified lipid peroxidation in mechanically separated red meat. They recommend optimising high-pressure conditions to limit lipid peroxidation in mechanically separated red meat [84,170].

Prooxidant activity in mechanically separated poultry. In the poultry industry, mechanical separation is employed for a more complete removal of muscle tissue from bones when manual extraction is impractical. The process of mechanically separating poultry results in significant changes to lipids and proteins [142,171], leading to the formation of an unpleasant flavour and diminished functional characteristics in the resulting mechanically separated meat. The use of extremely high pressure and aeration during processing reduces the oxidative stability of mechanically separated poultry, which includes small bone particles, blood, and phospholipids derived from nervous tissue [21,142]. To mitigate these changes, Dawson and Gartner recommend minimising storage time before use and completely removing oxygen from the system [172,173].

At a pressure of 10.55 kg.cm^−2^, mechanically separated poultry with very low fat and very high iron content are obtained, while a pressure of 2.81 kg.cm^−2^ contributes to stronger lipid peroxidation [174].

Prooxidant activity in mechanically separated fish. Fish muscles are characterised by a specific structure divided into easily separable myotomes, allowing muscle tissue to be easily separated from bones and skin at relatively low pressures [175]. The breakdown of fish muscles during mechanical separation creates conditions for the development of lipid peroxidation [176]. A significant increase in 2-thiobarbituric acid reactive substances (TBARS) is observed in defrosting mechanically separated fish [177]. TBARS levels rise proportionally to freezing time, with internal organs, spine bones, and dark fish muscles being the most susceptible to lipid peroxidation [153].

The oxidative stability during storage of mechanically separated meat from Longnose sucker (*Catostomus catostomus*) and White sucker (*Catostomus commersoni*), obtained during different seasons, varies [178]. The concentration of non-heme iron in frozen mechanically separated fish increases during storage [176]. In mechanically separated Atlantic cod (*Gadus morhua*), the content of non-heme iron is about 50%, and in mackerel (*Scomber scombrus*), it ranges from 20% to 64% of the total iron [179]. This increase is more pronounced at storage temperatures of −14 °C compared to −20 °C and −40 °C.

The prooxidant effect of high-pressure treatment. High-pressure treatment is reported to effectively inactivate microorganisms in meat [180]. Microorganisms have been found to be efficiently inactivated at pressures higher than 400 MPa. This pressure level is critical for initiating lipid peroxidation and is associated with the concentrations of heme iron in the meat. Guyon et al. [180] reported a close relationship between lipid and protein oxidation. It is proposed to control the entire technological process, from raw materials through the finished product, to storage and consumption [180].

The prooxidant effect of salting. Sodium chloride also influences the oxidative stability of muscle lipids [13]. It alters the ion power of the solution, a well-known activator of enzyme oxidation activity in meat [35]. The activity of antioxidant enzymes in pork muscles decreases proportionally as the concentration of sodium chloride increases from 0.5% to 2.0% [150]. This indicates that the reduced oxidative stability of salted meat can be attributed to the decreased activity of endogenous antioxidant enzymes. The addition of sodium chloride and magnesium dichloride with an ionic force of 0.70 or 0.35 increases TBARS in both raw and baked minced pork stored at 4 °C and −20 °C [181]. Sodium chloride accelerates lipid peroxidation in meat subjected to autolysis (ripening) [182]. However, such an influence has not been observed for heat-treated meat and frozen saltwater fish [183].

Sodium chloride stimulates lipid peroxidation in muscle cells [184]. Salt releases iron from the haem proteins in the sarcoplasmic fraction of turkey muscles [28], enhancing the development of lipid peroxidation. This effect is attributed to the ability of chlorine ions to dissolve ferro and ferric ions and form complexes with them [185]. Such complexes are more reactive and catalyse lipid peroxidation more strongly than iron ions [186]. The impact of sodium chloride on muscle lipid peroxidation development depends on the amount of so-called “free” water [187]. The impurities present in cooking salt could be a possible cause of its prooxidant effect [50]. The use of encapsulated cooking salt is an effective technological approach to limit the catalytic effect of sodium chloride on lipid peroxidation in meat products [188].

The prooxidant effect of metals with variable valence. Iron, copper, cobalt, and nickel have been identified as possible prooxidants in muscle tissue [189,190,191,192]. Metals with variable valence, metal proteins, and enzymes are normal constituents of muscle tissue and meat as a whole [193]. Some of these metals initiate the lipid peroxidation of cell membranes [194]. Catalysts of enzyme lipid peroxidation and haemoglobin are natural components of blood. Therefore, if carcasses are adequately exsanguinated, they pose minimal problems for most muscle foods [195]. An exception is fish, where blood cannot be removed before cleaning, and the remaining blood initiates lipid peroxidation [196].

The catalytic activity of metals with variable valence in skeletal muscles is restrained by the formation of proteins such as ferritin, transferrin, myoglobin, serum albumin, and ceruloplasmin. When metals with variable valence are complexed with these proteins, their ability to initiate lipid peroxidation decreases as they are associated with their oxidised and less reactive forms [197,198]. This binding reduces the redox potential of the metals, and their position on the protein prevents them from reacting with lipids [199]. About 10% of metals in skeletal muscles are not associated with macromolecules [194]. Metals with low molecular weight are indicated to be catalytically active and capable of initiating lipid peroxidation [122].

During heat treatment, iron-containing proteins denature and release the so-called “free” or “non-heme” iron [200,201]. Myoglobin is the most likely source for the release of low molecular weight during muscle heat treatment, as ferritin is not susceptible to such reactions [202]. The “free” iron can bind to lipid membranes and thus be closer to the oxidised substrate [122,199]. Many thermally induced changes in iron distribution occur above a known temperature limit [203]. When the temperature of the processed meat product exceeds this limit, its oxidative stability significantly decreases [195,204,205]. Therefore, temperature control of the finished product may be an effective method to limit lipid peroxidation in thermally treated meat and fishery products [206].

The prooxidant effect of ionising radiation. Irradiation of meat results in the formation of free radicals that initiate lipid peroxidation [207]. The total number of microorganisms in meat and fishery products can be reduced below permissible norms with doses of less than 10 kGy [208]. However, when irradiated with 10 kGy, the sulphur-containing components of the meat are oxidised, leading to the development of a non-specific flavour [209]. Vacuum packaging, modified atmosphere packaging, or processing with antioxidants of irradiated meat products can help retain the development of such an unpleasant odour [208].

The formation of oxy and hydroxy fatty acids, as well as products of membrane and sarcoplasmic lipid peroxidation that are potentially harmful to human health, is observed with γ-rays treatment [210]. Despite this, treatment with γ-rays has been demonstrated to extend the shelf life of cured meat products while maintaining acceptable levels of lipid peroxidation [209,211].

### 6.4. Antioxidant Factors

The type of muscle lipids, the degree of unsaturation, and the presence of natural antioxidants affect the oxidative stability of meat [114,212]. Three strategies are discussed for modifying the antioxidant protective system in muscle tissue. These include the activation of the inherent protective antioxidant systems, the introduction of exogenous antioxidants through animal feed, and technological approaches involving the addition of exogenous antioxidant ingredients in muscle foods.

#### 6.4.1. Own Endogenous Antioxidant Systems

Lipid-soluble endogenous antioxidants. Muscles possess a multicomponent antioxidant protective system [114]. Endogenous antioxidants can be categorised as soluble in lipids, water (sarcoplasmic), and sarcoplasmic antioxidant enzymes. These antioxidants play a crucial role in limiting the action of prooxidants, removing free radicals, and deactivating reactive oxygen products [84].

α-tocopherol. The main natural antioxidant in muscles is α-tocopherol [114,131]. Other tocopherols such as α-, β-, γ-, and δ- are also present [84,213]. These compounds possess a phenolic structure that neutralises free radicals by forming an α-tocopherylquinone radical, which has low energy and does not contribute to the further development of lipid peroxidation processes [82]. Tocopherols exist in nature in four isomeric forms, with their antioxidant activity following the order δ- > γ- > β- > α- [114,214]. In meat products, mixtures of plant tocopherol isomers have been found to be more effective antioxidants than pure α-tocopherol [114,144].

Carotenoids constitute another group of lipid-soluble endogenous antioxidants in skeletal muscles [114]. These compounds are obtained through animal feed, and their antioxidant action is attributed to their ability to deactivate oxygen species [215]. Evidence suggests that carotenoids, including lutein, lycopene, β-kryptoxanthin, and astaxanthin [216,217] are more effective in suppressing peroxyl radicals (LOO·) of *β*-carotene and *α*- tocopherol [218].

Ubiquinone, also known as coenzyme Q, is a chemical compound with an isoprenoid lateral chain [214]. Present in mitochondria, it acts to inhibit lipid peroxidation by deactivating free radicals [218].

Sarcoplasmic water-soluble endogenous antioxidants. Sarcoplasmic water-soluble endogenous antioxidants play a crucial role in preventing lipid oxidation in beef, chicken, and pork, which have a significant potential for lipid oxidation [219,220]. This potential is attributed to low-molecular organic compounds present in muscle cells, such as protein derivatives produced during meat autolysis, along with various enzyme systems exhibiting antioxidant activity [221].

Carnosine, identified as (2S)-2-(3-aminopropanamido)-3-(3H-imidazol-4-yl) propanoic acid, and *anserine*, recognised as (Z)-N-(3-amino-1-hydroxypropylidene)-3-methyl-L-histidine, are endogenous muscle dipeptides. Higher concentrations of carnosine and anserine are found in both light and dark muscles [222,223,224]. Carnosine acts as a membrane protector, inhibiting lipid peroxidation by capturing copper ions [225]. As a muscle dipeptide, carnosine serves as a buffering agent similar to antioxidants [224,226]. Its antioxidant mechanism appears to involve a combination of its ability to function as both a donor and a free radical acceptor [227]. The hydrophilic nature of carnosine ensures prevention within the sarcoplasmic environment, where numerous free radicals and catalysts of lipid peroxidation are encountered [227]. Carnosine is effective against cupro and cupric ions but not ferric and ferrous ions [228]. It serves as a stabilising antioxidant factor in poultry [227].

Glutathione, a tripeptide, effectively suppresses lipid peroxidation by deactivating free radicals [82,228]. It provides a source of electrons that allows glutathione peroxidase to enzymatically break down hydrogen peroxide (H_2_O_2_) and lipid hydroperoxides [220,228,229].

Muscle membranes ceruloplasmin, which acts as a ferroxidase, catalyse the oxidation of ferrous to ferric ions and reduce oxygen to water. The membrane enzyme, initiated either by ADF–Fe^2+^ or non-enzymatically through the iron redox cycle, is regulated by ceruloplasmin. However, ceruloplasmin cannot suppress membrane lipid peroxidation inhibited by hydrogen-activated metmyoglobin [230,231].

The concentration of *L-ascorbic acid* determines whether it initiates or inhibits lipid peroxidation. At low concentrations, ascorbic acid initiates lipid peroxidation by converting iron into active ferrous ions, while at high concentrations, ascorbate inhibits lipid peroxidation by deactivating free radicals from the chain reaction [82]. Ascorbic acid is rapidly broken down during meat storage due to its oxidation by metals [232].

Polyamines, such as putrescine, spermidine, and spermine, present in almost all animal tissues, inhibit lipid peroxidation by deactivating free radicals and the catalytic reactions of iron [233]. The antioxidant activity of polyamines increases with the number of amino groups in their molecule (spermine > spermidine > putrescine), similar to their inhibitory effect on lipid peroxidation in vitro [234,235].

Urate, which contains an amino group, inhibits oxidation reactions [214] by binding iron and removing free radicals from the system [31,82].

Sarcoplasmic antioxidant enzymes. Among sarcoplasmic antioxidant enzymes, *catalase* is an iron-containing enzyme [31,236] that can inactivate hydrogen peroxide (H_2_O_2_) [82] by breaking it down into water and oxygen [237,238]. Glutathione peroxidase, found in numerous biological systems, controls the formation of hydroperoxides [34,239]. Unlike catalase, glutathione peroxidase responds to both lipids and hydrogen peroxide [36,240]. The activity of catalase and glutathione peroxidase varies depending on the muscle type in animals of the same biological species [236,240]. Glutathione peroxidase may inhibit lipid peroxidation initiated by ascorbate and iron [82] through a soluble, thermally labile factor. Atmospheric oxygen or ozone can undergo transformation into superoxyl anion radicals (O_2_^−^·) by gaining an electron [31,34]. Superoxyl anion radicals (O_2_^−^·) contribute to the development of lipid peroxidation by introducing prooxidant metals and forming conjugated fatty acids and perhydroxyl radicals (HOO·) from unsaturated fatty acids [241,242]. The heat treatment of meat up to 70 °C leads to complete deactivation of catalase and partial inactivation of glutathione peroxidase [243].

#### 6.4.2. Exogenous Antioxidants

Various exogenous antioxidants are employed to inhibit lipid peroxidation and prolong the shelf life of muscle foods [22]. Long before the chain nature of lipid peroxidation was established, substances were known that, when added in small amounts, suppressed the initiation of oxidative processes, and were termed antioxidants [82]. Antioxidants form stable products that interrupt the chain reaction [20]. Another group of compounds called synergists enhances the action of antioxidants, although they themselves lack antioxidant properties [244]. Synergists deactivate heavy metal ions by binding them in complexes, thereby inhibiting their prooxidant action [244]. It is recommended to introduce antioxidants into the system during the early stages of the induction period when there are no free radicals present yet [245].

Depending on their origin, antioxidants are classified into natural and synthetic categories [197,246]. Natural antioxidants include tocopherols [114,144], rosemary extracts [197,246,247], essential oils of spices [247,248], and others. Polyatomic phenols exhibit strong antioxidant action, and examples of such compounds include gallic acid and its derivatives, bio-phospholipids, flavonoids, aromatic amines, some sulphur-containing substances, etc. [244,249]. Known antioxidants are further divided into three main groups: strong antioxidants, weak antioxidants, and synergists [246,250,251]. The first group includes free radical scavengers, such as α-tocopherol or phenolic compounds incl. butylated hydroxytoluene (BHT) and butylated hydroxyanisole (BHA), rosmarinic acid, quercetin, and dihydroquercetin [252,253,254]. Their effect is due to their free hydroxyl groups [255].

Weak antioxidants form radicals (AO·) and the rate of oxidation suppresses but practically does not stop [256]. Chain oxidation reactions can be inhibited not only by increasing the rate of chain termination, as is the case with the action of phenols and some so-called “weak” antioxidants (quinones, amines) but also by reducing the rate of free radical formation through degenerate chain branching reactions [257]. Another group are metal chelators. They can slow lipid peroxidation by varying the reduction potential of the transition metals [258] and avoiding direct contact between transition metals and hydroperoxides [259]. Organic acids (e.g., citric acid), ethylenediaminetetraacetic acid (EDTA), polyphosphates, and proteins are common food chelators used in the food industry [260].

When two antioxidants with different mechanisms of action when used together, a strong synergistic effect occurs [246]. It is possible to use synergist compounds that do not have antioxidant properties or are very weak antioxidants. Examples of synergists are some polybasic organic acids, such as citric, tartaric, ascorbic, some amines, and inorganic acids, e.g., phosphoric acid or its acid esters [261]. If the formed compounds (intermolecular complexes, adducts, dimers, or phenolics) have a higher antioxidant activity it could retard the rancidity as a result of synergism [261]. A typical example is an L-ascorbic acid—tocopherol synergism. It may be applicable to muscle foods where oxidation occurs in the cell membrane [246].

Phenolic compounds, notably strong antioxidants, play a crucial role in antioxidant activity [144]. The effectiveness of phenols as antioxidants is based on their ability to deactivate free radicals and reduce the concentration of variable valence metals [231]. Their action involves interrupting the chain reaction by interacting with active radicals L· and LO· [211]. Following the principle of the indestructibility of free valences, new, inactive radicals of antioxidants are formed, preventing the continuation of the chain reaction [144,231]. The impact of phenolic antioxidants is attributed to their free hydroxyl groups [252]. Electron donor groups, such as methyl, methoxyl, etc., in the o- and p-positions significantly enhance antioxidant activity, while electron acceptor groups (nitroso-, carboxyl, etc.) diminish it [252,255]. An “inversion” in their action occurs above a certain concentration, explained by the increased rate of the breakdown of free radical hydroperoxides under the antioxidant influence [174]. Consequently, the concentration of phenolic antioxidants typically does not exceed certain limits (in tocopherols, 0.02–0.03%) [231,253]. A notable property of polyphenols is their ability to maintain effectiveness at high temperatures [247]. Phenolic antioxidants are found in various spices and herbs such as rosemary, cinnamon, black pepper, nutmeg, liquorice, anise, cassia bark, fennel, prickly ash, round cardamom, basil, black coriander, and ginger [145,253,254,255] and plant extracts like grape pomace [145,262], basil [263,264], sage [265,266], oregano [266], thyme [265,267], and artemisia [268]

Rosemary (*Rosmarinus officinalis* L.) is widely used, with extracts at a concentration of 0.02–0.05 g.kg^−1^ inhibiting lipid peroxidation in beef, pork, turkey, chicken, sausages, and herring fillet [197,250,253]. The phenolic ingredients responsible for rosemary’s antioxidant activity include carnosol, rosmanol, rosmaridiphenol, carnosic acid, rosmarinic acid, etc. [144,197].

The use of grape by-product extracts and their bioactive compounds as natural antioxidants in meat products was discussed. The grape pomace extracts were shown to be suitable for inhibiting lipid oxidation and for quality preservation of muscle foods [269]. Sáyago-Ayerdi et al. [262] suggested grape pomace concentrate could be applied as a dietary supplement to chicken feed. It was discovered that grape pomace could be an effective inhibitor of lipid oxidation of chicken patties both chilled and long-term frozen stored. Similarly, Bennato et al. [270] found the dietary supplementation of the chicken’s diet with 7% grape pomace inhibited lipid oxidation and decreased volatile aldehydes content after 7 days of storage of raw breast chicken meat. Another study [271] found there were no significant differences between the percentage of lipid peroxidation inhibition in ex vivo and in vivo experiments when the effect of 6% grape pomace supplementation in the broiler diet was investigated. Goñi et al. [272] suggested 30 g/kg grape pomace in combination with 200 mg/kg α-tocopheryl acetate (vitamin E) to be included in a corn–soybean staple diet for chickens. It has been found such a diet can significantly reduce lipid oxidation of refrigerated chicken meat. In addition, Chamorro et al. [273] found the dietary treatment of broiler chicks with 10% grape pomace reached the protective effect of α-tocopherol by reducing the susceptibility of meat to lipid oxidation and increasing the PUFA content.

The study by Guerra-Rivas et al. [274] demonstrated that dietary treatment with 50 mg/kg grape seed extract and 5% dried red grape pomace inhibit the oxidation of sliced, packaged, and under modified atmosphere (80 O_2_/20%CO_2_) lamb m. *Longissimus thoracis et lumborum* were stored for 14 days in retail conditions but were not so effective in preventing sensory spoilage and shelf life of lamb meat in comparison with dietary treatment with 500 mg/kg vitamin E (α-tocopheryl acetate). The dietary wine grape pomace supplementation is an effective manner for increasing the antioxidative activity of lamb meat because a decrease in reactive oxygen species and malondialdehyde levels induced the lamb’s m. *Longissimus dorsi* [275].

Several in vitro studies have demonstrated the effectiveness of grape pomace supplementation on lipid peroxidation inhibition in meat products. Garrido et al. [276] found two types of red grape pomace extracts can improve oxidative and colour stability, and finally to guarantee the best global acceptability of 6 days stored pork burgers. Carpes et al. [277] evaluated the effect of spray-dried and lyophilised powders made from winery by-products on the oxidative stability of chicken pâté. Two pâté formulations containing lyophilised and microencapsulated grape pomace were produced during refrigerated storage (4 °C/42 days). It was found that the addition of both natural antioxidants in chicken pâté resulted in lower TBARS values than in pâté. Sampaio et al. [278] have suggested the use of natural antioxidant combinations such as sage, oregano, and honey. It has been found that this combination is capable of protecting cooked chicken meat from lipid oxidation during 96 h storage at 4 °C. Treatments with oregano + sage + 5% or + 10% honey minimised hexanal levels and contributed to establishing only traces of free 25-OH, 7-k, 7α-OH, and 7β-OH cholesterol oxides.

Basil and its extracts are also widely discussed substances that are used in vitro as potential inhibitors of lipid oxidation in muscle foods. The in vitro technique of adding basil extracts to various types of meat and meat products has been applied as an approach to inhibit the lipid oxidation processes. Dimitrov et al. [266] determined the reduction in the MDA content in poultry meat treated with aqueous extracts of wild basil (*Clinopodium vulgare* L.) compared to nontreated poultry. The treatment with an ethanolic extract of basil was applied to fresh chicken minced meat and fresh and cooked pork patties [264]. A significant reduction in both POV and TBARS was found in refrigerated and chilled pork patties pretreated with NaCl [270]. For the purpose of improving the colour and lipid oxidative stability, Falowo et al. [279] successfully used 2% and 4% sweet basil (*Ocimum basilicum* L.) essential oil to aerobically packaged minced beef stored 7 days at 4 ± 1 °C. Falowo et al. [280] found thirty-two bioactive compounds with antioxidant and antimicrobial action. Substances mainly included estragole—41.40%; 1,6-octadien-3-ol, 3,7-dimethyl—29.49%, and trans-α-bergamotene—5.32%. Similar experiments to the in vitro addition of basil (*Ocimum basilicum* L.) essential oil in beef burgers were conducted by Sharafati-Chaleshtori et al. [281]. This research shows no significant differences between different concentrations of added basil essential oil in regard to lipid oxidation decreasing in raw beef burgers.

Juntachote et al. [279] have found that dried holy basil powder is a more effective inhibitor of lipid oxidation in cooked ground pork stored for 14 days at 5 °C compared to its ethanol extracts and ethanol extracts of galangal. It has been established that the antioxidant effectiveness of dried holy basil powder was smaller compared to the blend of citric acid/ascorbic acid/α-tocopherol added to cooked ground pork in concentrations 0.3%/0.5%/0.02% [263]. Finally, Cichoski et al. [282] found antioxidant activity in terms of the lipid fraction when the inside of an Italian-type salami was treated with 0.75 mg/g basil (*Ocimum basilicum* L.) essential oil during processing and storage. No similar activity was found for the protein fraction.

Sage is another well-known natural antioxidant used to limit lipid peroxidation processes in muscle foods [265]. Antioxidant activities of sage extracts were studied from the middle of the 90th years of the 20th century [283]. The addition of 30 mg/kg sage extracts and vitamin E in a model meat system of cooked beef homogenate effectively reduced 53% MDA levels [283]. According to Tanabe et al. [265] liquid sage extract, like the sansho and ginger extracts, showed the strongest inhibition of lipid oxidation in cooked pork homogenate when compared to 22 culinary herbs and spices. Dietary-applied sage oil extracts (500 mg/kg) were found to effectively inhibit lipid oxidation in broiler meat [284]. Several in vitro experiments have also demonstrated the antioxidant activity of sage in various meat products. The delayed formation of lipid oxidation-derived products through 9 days of storage of the turkey meatballs packed in a modified atmosphere (80% O_2_/20% CO_2_) stored at 4 °C when sage extracts (0.02% and 0.05%) were added was determined [285]. Bak et al. [267] are of the opinion that all 15 examined sage (*Salvia* spp. Labiatae) extracts were capable of reducing lipid oxidation in cooked meat. The addition of 3% w/w sage essential oil was found to significantly reduce the lipid oxidation of raw and cooked at 85 °C for 30 min ground pork and beef, stored for 12 days at 4 °C [286]. Comparatively higher antioxidant activity and L* values when sage essential oil was added to cooked minced beef were reported too [287]. The addition of 1 g sage/kg cooked chicken effectively controlled lipid and cholesterol oxidation [288]. It minimises the pro-oxidant effects of salt, cooking, and storage. The same study found that in fresh raw chicken (on day 0) only the amount of 7-ketocholesterol was higher. After 30 days of storage, the formation of 7β- and 7α-hydroxycholesterol was found [288]. In their study, Bianchin et al. [289] also demonstrate that when the lyophilised sage extracts were added to poultry pátê the lipid oxidation was inhibited during storage.

The strategy for dietary application of oregano and its extracts as antioxidants in livestock feeding was discovered in many studies. As an approach to improvement of the oxidative stability and consumer acceptance of poultry meat, Forte et al. [290] have considered the dietary 0.2 g/kg oregano (*Origanum vulgare* L.) aqueous extract addition to the broiler’s feed enriched with CLA and n-3 PUFA. It was found that dietary oregano extracts influenced antioxidant capacity, reduced the thiobarbituric acid-reactive substances, and improved the poultry resistance to oxidation. Botsoglou et al. [291] used a similar strategy to improve the oxidative stability of raw and cooked chicken breast and thigh meat stored for 9 days at 4 °C. Dietary supplementation of 50 or 100 mg of oregano essential oil/kg forage was found to have an antioxidant effect inferior to that of dietary supplementation of 200 mg α-tocopheryl acetate/kg. The results reported by Marcincak et al. [292] regarding poultry chicken breast and especially thigh meat stored frozen for 12 months at −21 °C are similar. The diets containing 0.5% and 1% dried oregano powder [293] were demonstrated to reinforce the antioxidant enzyme activity of glutathione peroxidase and superoxide dismutase and could reduce TBARS values in 5 d stored chilled duck’s breast.

Simitzis et al. [294] reached similar conclusions when they supplemented the diet of lambs with 1 mL/kg of oregano essential oil. They found delayed lipid oxidation (MDA formation) during storage of both chilled and long-term frozen lamb. However, compared with α-tocopheryl acetate supplementation, dietary supplementation at the level of 100 mg oregano essential oil/kg feed was less effective in suppressing the lipid oxidation of frozen at −20 °C chicken breast and thigh muscle meat during a 9-month storage period [295], and in turkey breast and thigh minced meat [296].

Post mortem in vitro pre-treatment with 5 g/kg oregano extract (*Origani vulgaris herba* L.) extract to the minced turkey meat patties [296], refrigerated raw poultry meat [266], and ground trimmed beef [297] also showed lower values for MDA content, improved the oxidative stability, and stabilised the redness meat rating stored 12 days at 4 °C. Similar results are reported on raw and cooked at 85 °C for 30 min ground pork and beef [286] when 3% w/w oregano essential oil was added and on cooked minced beef [287].

According to the findings of Manhani et al. [298], the in vitro addition of oregano extract is not as effective in lowering TBARS as the addition of sodium erythorbate or deodorised rosemary extract in precooked beef burgers.

The dietary supplementation of 0.5% thyme (*Thymus vulgaris*) broiler chickens under heat stress [299], 100 mg/kg thyme extract to the basic diet of Japanese quails [300], and 3% thyme to the diet of growing rabbits [301] was published. Those studies demonstrated the antioxidant activity of thyme and found significantly lower malondialdehyde (MDA) concentrations in the broiler chicken thigh muscle [299], Japanese quail thigh meat [300] and raw and freeze-dried rabbit meat [301].

Natural antioxidants such as thyme minimize the oxidative changes in red meat and meat products [302]. The antioxidant activity of thyme added to pork homogenate significantly suppressed lipid oxidation expressed as a thiobarbituric acid (TBA) value [265].

In vitro supplemented 0.2% thyme extract (*Thymus schimperi*) significantly improved the meat’s oxidative stability and guaranteed 3 weeks of its shelf life [303]. Similar conclusions were made for the in vitro addition of 0.9% thyme essential oils in minced pork stored for 15 days at 3 °C [304]. Boskovic et al. [304] found significantly lower oxidation and lipolysis levels in vacuum-packaged minced pork than in a modified atmosphere (MAP) (30%O_2_/50%CO_2_/20%N_2_). Hęś et al. [305] reported a significant decrease in lipid oxidation and preserved the methionine and lysine availability and protein digestibility in frozen (at −18 °C) stored for 6 months of fried pork meatballs, enriched with the extracts from thyme. A significant reduction in TBARS (36.58–46.34%) and metmyoglobin (16.25–18.47%) in 2 months frozen and stored at −18 °C beef burgers incorporated with Shirazi extract was discussed too [306]. Similar results were found after 6 months of frozen storage at −18 °C of raw and cooked heat-treated pork patties enriched with dried thyme or its ethanol extract [264].

Another representative of plants rich in phenolic antioxidant compounds is *Artemisia* spp. Extracts from these plants are mainly applied as dietary supplements to chicken feed or in vitro in chicken meat products.

Kostadinović et al. [307] suggest the inclusion of 200 g/kg *Artemisia absinthium* as a dietary additive to feed broiler chickens to manage the antioxidant status and quality of fresh chicken meat. Those authors found a significant decrease in the concentration of malondialdehyde (MDA) in plasma in comparison with the control group, accompanied by significantly higher glutathione peroxidase activities in the blood of chickens.

An experiment conducted by Wan et al. [308] evaluating the effect of including 1.0 g/kg of enzyme-treated *Artemisia annua* L. in broiler diets found that after 15 days of storage in chicken breast and thigh muscles at 4 °C, malondialdehyde concentration significantly decreases.

Similar conclusions were drawn by Panda et al. [268], where the authors fed broiler chickens a feed supplement of 2% dried *Artemisia annua*; by Rahiminiat et al. [309] who added *Artemisia sieberi* essential oil to the diet of broiler chickens; by Kim et al. [310] investigated dietary supplementation of 2.5 or 5.0 g/kg Lactobacillus-fermented *Artemisia princeps* in male Hy-line Brown chickens; and by Cherian et al. [311], who studied the effect of feeding a supplement of 2% or 4% dried *Artemisia annua* to broiler chickens.

Contrary to the data obtained in broiler chickens, no statistically reliable effect on the antioxidant status of the meat was demonstrated in lambs receiving oral administration of 400 mg/kg *Artemisia herba alba* essential oils. Aouadi et al. [312] found that lipid oxidation and the colour of lamb muscles were not affected by feed supplementation with *Artemisia herba alba* essential oils after 7 days of aerobic storage.

As a result of in vitro experiments investigating the inhibition of lipid oxidation of raw chicken patties by adding an antioxidant combination of 0.05% L-ascorbic acid + 0.2% extract of Ganghwayakssuk (*Artemisia princeps* Pamp.) Hwang et al. [313] have established the most effective decrease in the TBARS, conjugated dienes, and peroxide formation. These results allowed the researchers to suggest that the addition of a combination of antioxidants successfully reduced the oxidative stress of raw chicken patties during 12-day refrigerated storage. Hwang et al. [313] concluded that the shelf life of chicken patties could be extended under these conditions. Choi et al. [314] also demonstrated that three concentrations (0.05, 0.1, and 0.2%) of Ganghwayakssuk (*Artemisia princeps* Pamp.) extract prevented lipid oxidation in raw chicken nuggets batter after 10 d of storage at 4 °C.

Finally, Falowo et al. [280] have established the significant inhibition of lipid oxidation in cooked ground pork during its 7-day storage at 4 °C, when the 2 mL/kg *Artemisia afra* essential oil was added.

Chain oxidation reactions can be inhibited not only by increasing the rate of chain termination, as is the case with the action of phenolics and some so-called “weak” antioxidants (amines, quinones), but also by reducing the rate of free radical formation through degenerate chain branching reactions [114]. This can be completed by introducing into the oxidising environment substances capable of reacting with hydroperoxides without the formation of free radicals [114]. Thus, hydroperoxides readily react with dialkyl sulphides. As a result, a sulfoxide is formed, capable of re-interacting with hydroperoxides. The product of this secondary reaction is a sulfone, which is formed at a much slower rate [114].

The mechanism of antioxidant action of this type of antioxidants is not chained, their effectiveness is significantly less than that of radical antioxidants. However, adding an oxidising system together with phenolic antioxidants can significantly increase their effect [315]. Some free acids and bases have a similar effect, causing the decomposition of hydroperoxides by an ionic mechanism without the formation of free radicals [114].

Amines. Amines exhibit two proposed mechanisms for inhibiting the chain radical processes of lipid peroxidation [114]. There are two opinions regarding the mechanism of action of amines as inhibitors of chain radical reactions of lipid peroxidation. According to one of them, the inhibitory action of amines is due to the donation of a mobile hydrogen atom from their molecule to RO_2_· radicals, in which non-reactive compounds are formed [315]. The least mobile atom in the amine molecule is the hydrogen from the amino group. According to the second opinion, in the process of inhibition, as an intermediate, a complex radical formed by the addition of RO_2_· is involved radical to the amine molecule, at the expense of a pair of free electrons from the nitrogen atom [114]. The first involves the donation of a mobile hydrogen atom from the amine molecule to LOO· radicals, resulting in the formation of non-reactive compounds. The second suggests that in the inhibition process, a complex radical is formed as an intermediate compound when the LOO· radical is added to the amine molecule, with the transfer of a pair of electrons from the nitrogen atom [315].

Quinones, possessing the capability to inhibit lipid peroxidation processes, attach to free radicals through their multiple connections. Weaker antioxidants form radicals that participate in another type of reaction. As a result, lipid peroxidation is inhibited but not completely halted [255].

Diacyl sulphides. Diacyl sulphides can inhibit chain radical oxidation reactions not only by accelerating the interruption of reaction chains but also by slowing down the formation of free radicals through degenerate chain branching reactions. These substances have the ability to react with hydroperoxides in the oxidised substrate without generating free radicals, forming sulfoxide. The resulting sulfoxide can then react with hydroperoxides, producing a sulfone at a much slower rate [316].

The combined use of two antioxidants with different mechanisms of action demonstrates a strong synergistic effect [144]. Synergistic effects are enhanced when the two inhibitors are used to act as peroxide scavengers or chelators, facilitating the interaction between free radicals and thus terminating the chain reaction [20,254,317]. Synergists can also include compounds that lack antioxidant properties or exhibit very weak antioxidant activity [318,319,320]. Certain amino acids, polyphosphates, cephalic, sulfhydryl, and other compounds act as active synergists in inhibiting lipid peroxidation in meat and fish products [321,322,323,324].

Complex formers. Complex formers play a role in inhibiting lipid peroxidation in muscle foods by suppressing the activity of transition valence metals. These substances have the ability to form durable chelating complexes with metals of variable valence, deactivating them as active prooxidants [174]. They indirectly contribute to a synergistic effect on lipid peroxidation [322,323,324,325,326]. Certain substances, such as citric acid, tartaric acid, and ascorbic acid, act as deactivators of metals while also serving as synergists [172,327,328].

The antioxidant action of *L-ascorbic acid* can be attributed to its role as a source of mobile hydrogen [144,174]. L-ascorbic acid can restore its oxidised and dehydrogenated form easily, facilitating regeneration and oxidation [172]. Polyhydroxy derivatives with appropriately placed carboxyl and hydroxyl groups are also effective in forming chelated complexes [327,328]. Many highly effective antioxidants are mixtures of several carefully selected antioxidants, synergists, and metal deactivators [172,327,328]. However, their efficacy is often substrate-specific, influenced by the composition of the oxidised substrate [329].

Some acids and bases also have inhibitory effects on lipid peroxidation by breaking down hydroperoxides through an ionic mechanism without forming free radicals. An example of such a compound is *benzoic acid* [330]. Commonly used complexes in meat and fishery products include *citric acid* and its salts, as well as *phosphates*, with *polyphosphates* being particularly effective antioxidants in heat-treated meat and fishery products, inhibiting up to 85% of lipid peroxidation [187]. In cured meat products, polyphosphates are less effective, suppressing 8–60% of lipid peroxidation compared to raw beef [321].

Other antioxidants. Indeed, some protein sources, such as *whey protein*, have been found to suppress lipid peroxidation in meat. Whey protein not only stabilises the emulsion of cooked sausages but also inhibits the development of off-flavours in muscle foods [331]. This suggests that proteins can contribute to the oxidative stability of meat products, potentially through their interactions with lipids and other oxidative processes.

Nitrites. Nitrites, particularly sodium nitrite, were commonly used in the meat industry for their various properties, including colour formation, preservation, and antioxidant effects [332]. When added to cured meat, sodium nitrite interacts with other compounds to form nitroso myoglobin or S-nitrosothiols [333]. These complexes exhibit antioxidant properties [334]. Nitric oxide, generated from nitrites, has several mechanisms that contribute to its antioxidant effects [335]:(1)Inhibition of Fenton’s reaction: nitric oxide can inhibit Fenton’s reaction, which involves the formation of ferrous ions and contributes to oxidative stress [335];

Suppression of enzyme activities: nitric oxide can suppress the activities of enzymes such as lipoxygenase and cyclooxygenase, which are involved in lipid oxidation processes [334,336,337];

(1)Interaction with iron: nitric oxide can interact with both non-heme and heme iron, preventing these metals from catalysing oxidative reactions; [336];(2)Radical acceptance: nitrogen oxide, nitrogen–oxide complexes, and S-nitrosothiols formed from nitrites act as radical acceptors, neutralising free radicals and interrupting chain reactions [338];(3)Protection of porphyrin: nitrogen oxide complexes with haem proteins protect porphyrin from releasing iron when exposed to hydrogen peroxide and hydroperoxides [339];(4)Stabilisation of lipids: nitrogen oxides formed outside the membranes during the smoking of meat products can stabilise unsaturated lipids [340].

Overall, the antioxidant effect of nitric oxide is multifaceted and contributes to the oxidative stability of meat products during processing and storage [341].

#### 6.4.3. Technological Methods for Inhibition of Lipid Peroxidation in Muscle Foods

Types of packaging. The choice of packaging plays a crucial role in minimising the processes of lipid peroxidation [144], and different methods are recommended [166] to limit the access of oxygen, a key factor in oxidative reactions:(1)Vacuum packaging is effective in inhibiting lipid peroxidation in both meat and fish [144,151]. The process should be carried out promptly after heat treatment to maximise the positive effects [156];(2)Modified atmosphere packaging (MAP) involves changing the composition of the air inside the package to slow down oxidative reactions. Fresh sea fish, for example, benefit from MAP. While MAP slows down lipid peroxidation compared to storage in the air, it may result in higher levels of TBARS compared to vacuum packaging.

The packaging technique plays a crucial role in maintaining the oxidative stability of mature salted anchovies [341] and beef [342] during cold storage. Poultry meat retains its inherent qualities when vacuum-packed [151]. Lipid peroxidation in baked broiler meat, particularly with an increased content of α-linolenic acid [161], and slides of chilled baked chicken breasts [156], is inhibited when subjected to vacuum packaging. MAP is recommended for fresh sea fish [343]. Variations in packaging materials significantly affect the limitation of lipid peroxidation in fish sausages. The sensory attributes of frozen and chilled meat are best preserved through vacuum packaging [157,166]. When utilising a modified atmosphere like carbon dioxide or nitrogen, lipid peroxidation progresses more slowly than in air storage, but TBARS levels are higher than in vacuum-packaged meat [155,157]. The presence of carbon dioxide in the atmosphere restricts lipid peroxidation processes [144]. At −10 °C, lipids exhibit no signs of rancidity for up to 12 months [163]. Smoked chicken legs packaged in an MAP can be stored for 25 days at 4 °C [344]. MAP contributes to maintaining the colour, taste, and flavour while suppressing microbial growth during the storage of chilled beef cuts [157,345], beef [157], ground beef [159], rabbit meat [346], roasted pork, and turkey meat [347]. To enhance the effects of packaging in a modified atmosphere and the inhibition of lipid peroxidation, Balev et al. [346,347] suggest treating chilled beef with natural antioxidants. To boost oxidation stability, MAP pork grill sausages with the addition of rosemary, ascorbic acid, sodium lactate, and red sugar beet roots were combined [348].

Edible films and coatings. Over the past decade, researchers have dedicated their attention to exploring environmentally friendly solutions for packaging meat and fish. Proposals for innovative strategies involve the application of environmentally friendly, biodegradable, and edible biopolymer films and coatings to meat, poultry, and fish products [349]. The primary objective is to restrict oxygen access and mitigate microbial cross-contamination in muscle foods [350]. Polysaccharides such as cellulose, carrageenan, chitosan, pectin, starches, gums, and alginates, as well as proteins like milk, collagen, and soy, have been discussed for this purpose. Additionally, various lipids, including essential oils, waxes, plasticisers, emulsifiers, and resins, have been considered [351].

Another innovative approach involves enriching edible films and coatings with active components to extend shelf life, reduce moisture evaporation losses, limit pathogenic microorganism growth, and slow down the development of putrefactive microbial spoilage. Notably, this approach aims to inhibit oxidative processes in the lipids, proteins, and pigments of muscle foods [352]. Examples of composite films for food packaging include those containing grass carp collagen, chitosan, and lemon essential oil [353], collagen and carboxymethyl cellulose films enriched with Boxthom barberry (*Berberis lyceum*) root extract [354], and films made from collagen from tuna skin, chitosan, and ultrasound-modified polyphenols from pomegranate (*Punica granatum* L.) [355].

Shokraneh et al. [356] have identified the effects of collagen fibres and green tea extract on the quality of vacuum-packed sausages. Typically, such edible coatings and films are frequently discussed in the context of preserving fish freshness. Various fish species have been reported to benefit from different edible coatings and films. Examples include treating with a gelatine film with propolis extract [357], covering red sea bream (*Pagrus major*) with a composite coating of chitosan and collagen derived from the skin of the blue shark [358], treating fresh salmon fillet (*Salmo salar*) with a film containing salmon bone gelatine, chitosan, gallic acid, and clove oil [359], or applying an edible alginate coating and surface treatment with dry distilled pink petal extract or L-ascorbic acid to paddlefish (*Polyodon spathula* Walbaum, 1792) [360].

Additional methods involve covering carp (*Cyprinus carpio*) fillets with antioxidants extracted from nutmeg, rosemary, thyme, ginger, marjoram, parsley, turmeric, basil, and ginger using water and ethanol [361]. There is also the application of biodegradable antibacterial, antioxidant, and pH-sensitive hydrogel films containing carboxymethyl cellulose, collagen from dried fish bladder, eucalyptus extract, or quercetin [362], and the use of an edible gelatine coating and *Portulaca oleracea* extract on fish sausages [363].

Glazing of frozen fish. The glazing of frozen fish has been applied to restrict oxygen access, prevent microbial contamination, and minimise moisture evaporation from the surface of frozen fish. In recent years, there has been a proposal to enhance frozen fish by incorporating natural antioxidant extracts. The quality of frozen bonito (*Sarda sarda*) fillets can be effectively maintained by glazing with sage extract (*Salvia officinalis*) [364]. Frozen curimbata (*Prochilodus lineatus*) fillets are glazed with water containing turmeric extract [365]. Glazing frozen Atlantic horse mackerel (*Trachurus trachurus*) involves the use of water containing octopus products [366]. Squid (*Pholidoteuthis massyae*) is glazed with water containing preservatives [367]. Frozen Atlantic mackerel (*Scomber scombrus*) benefits from glazing with water containing quinoa [368]. Glazing frozen bigeye tuna (*Thunnus obesus*) entails the use of water containing rosemary acid, bamboo leaves, and sodium lactate [369]. Finally, frozen Nile tilapia (*Oreochromis niloticus*) is glazed with water containing two types of Kaua’i pricklyash (*Zanthoxylum kauaense*) [370].

Vacuum impregnation of muscle foods. A novel approach to inhibit lipid peroxidation in muscle foods involves vacuum impregnation at various pressures [371]. Demir et al. [372] discovered that employing vacuum impregnation as a pre-treatment for traditionally marinated beef m. *Longissimus dorsi* with onion juice improved meat tenderness by 28.25%, with no alteration in colour brightness (L*). Simultaneously, tyramine concentration decreased, and TBARS levels were significantly lower than the limit value of 0.58 mg MDA.kg^−1^, as indicated by Martinez et al. [373], beyond which a rancid taste and smell are established.

Vacuum impregnation is a more effective method for salting Russian sturgeon fillets compared to salting under atmospheric pressure [374]. Fillets subjected to vacuum impregnation exhibited lower levels of TVB-N (total volatile basic nitrogen) and protein carbonyls, measuring 15.91 mg.100 g^−1^ and 311.38 nmol.L^−1^, respectively, after 5 h of salting.

Zhao et al. [375] demonstrated the preservation of better quality and inhibition of lipid peroxidation in vacuum-impregnated grass carp (*Ctenopharyngodon idella*) fillets covered with an edible coating of chitosan infused with three types of water-soluble polyphenolic extracts—pomegranate bark, grape seeds, and green tea. Vacuum impregnation proved most effective in fillets covered with an edible film infused with green tea extract, packed in sterile polyethylene bags in an air atmosphere. The TBARS in this sample increased from 0.35 to approximately 1.70 mg MDA.kg^−1^ after 12 days of storage at 4 °C.

Furthermore, Zhao et al. [376] investigated the potential inhibition of protein oxidation by combining an edible coating of fish gelatine and grape seed extract with vacuum impregnation of tilapia (*Oreochromis niloticus*) fillets stored for 12 days at 4 °C. This combination was found to retard protein oxidation. The observed data are attributed to the formation of disulphide bonds, reducing total sulfhydryl groups and lowering Ca^2+^-ATPase activity.

## 7. The Quality of Muscle Foods Affected by Oxidative Processes

### 7.1. Effect of Muscle Fibre Type on Lipid Peroxidation-Induced Alterations in Pork Leg Ham Flavour

Volatile aromatic compounds of various types have been identified in ham produced from different pig breeds, including Landrace × Large White Pig, Douros × Gascon-Meychan, Pietran × Gascon-Meychan, and Large White Pig × Gascon-Meychan [377]. Each pig breed has exhibited the isolation of 7 to 42 flavour compounds individually [378]. It has been established that 1-octen-3-ol, 2,3-butanedione, and acetoin exert the most significant influence on ham flavour. Levels of 1-octen-3-ol, a secondary product of unsaturated fatty acid oxidation, were observed to be lower in fat-rich thighs characterised by a higher proportion of saturated lipids. This occurrence is likely attributed to the lower ratio of phospholipids to triacylglycerols [379].

The impact of muscle fibre type and breed on the palatability of pork products remains a topic of discussion. Deng et al. [311] reported no significant reciprocal influence between the histological structure and sensory properties of mature pork leg ham. However, a notable correlation was identified between the number of muscle fibres of type II b, the pale colour, and the ham’s maturity [312]. It is probable that anabolic steroids and other growth-promoting substances accelerate the accumulation of muscle proteins and, to a lesser extent, fat [380].

### 7.2. The Aroma of Roast Meat Related to the Maillard Reaction Affected by Lipid Peroxidation

Oxidised phospholipids are identified as the primary precursors of volatile aromatic compounds responsible for the development of unpleasant meat aroma. In contrast, the Maillard reaction is regarded as the predominant chemical mechanism leading to the formation of aromatic volatile compounds with an aroma in roasted meat [381]. Through Maillard reactions, a diverse array of heterocyclic compounds is generated, contributing to the distinctive aroma of roasted meat. These compounds encompass O-, N-, and S-heterocycles, such as furans, furanones, pyrazines, oxazoles, thiazolinos, thiophenes, and cyclic polysulfides [382]. Sulphur-containing heterocyclic aromatic compounds play a crucial role in the aroma profile of roasted meat. While most of these compounds are formed in small concentrations, they possess a very low sensory threshold [381].

Oxidised products of unsaturated fatty acids emerge in the presence of phospholipids, unsaturated fatty acids, and/or nonpolar phospholipid heads [383]. Simultaneously, the quantities and proportions of sulphur-containing heterocyclic compounds notably decrease, whereas the furans content remains relatively unchanged [382]. A plausible mechanism for the reduction in sulphur-containing volatile heterocyclic compounds involves the reaction between aldehydes derived from fatty acid oxidation and hydrogen sulphide. H_2_S diminishes the capacity to form sulphur-containing heterocyclic compounds. Another potential mechanism entails reactions between furfural (a by-product of the reaction between reducing sugars and amino groups of amino acids) and a polar head of a phospholipid molecule like ethanolamine [384]. Similarly, aldehydes (secondary derivatives of meat lipid peroxidation) can react with ammonia, resulting in the formation of non-volatile Schiff bases due to ammonia’s reduced ability to synthesise pyrazines and alkylpyrazines [385]. Oxidised components of unsaturated fatty acids, including 1-octen-3-ol, hexanal, and pentanol, exhibit different relative proportions compared to those observed when unsaturated fatty acids are heated alone. This supports the hypothesis of the involvement of Maillard reaction products in the oxidation of fatty acids [382].

Extracts of volatile aromatic products in roasted meat have revealed the identification of over 1000 chemical compounds. Of these, a maximum of 32 could be attributed to the interaction between Maillard reaction derivatives and lipid oxidation—23 in beef, 5 in goat, and none in pork [381,383]. The interaction between lipids and the Maillard reaction can be summarised as suggested in Figure 4.

Muscle lipids can engage in a Maillard reaction with amino groups of the polar head of phospholipids or with certain aldehydes, which are oxidised secondary breakdown products of fatty acids [381,383]. Oxidised lipid derivatives may participate in Strecker degradation and the Maillard reaction during the processing of meat or fish, leading to the formation of a diverse range of volatile compounds [386]. Lipids play a role in the development of meat flavour by reducing the levels of sulphur-containing compounds and providing volatiles such as carbonyls or alcohols [387]. The positive impact of phospholipids on meat flavour is associated with the suppression of the Maillard reaction, particularly in the formation of sulphur-containing volatile compounds [388]. This suppression effect becomes more pronounced when the meat contains higher amounts of polyunsaturated fatty acids (PUFA), as observed in the case of chicken [382,384].

### 7.3. Effect of Derivatives of Lipid Hydroperoxide Degradation on Meat Aroma

Lipids play a direct role in the formation of aroma in meat and fish through interactions with other components or through autoxidation during processing, cooking, and storage. The impact is more pronounced when muscle lipids are more unsaturated [388]. The oxidative changes in lipids result in a deterioration of sensory properties in meat and fish products, leading to reduced consumer acceptability [84]. Aroma, being a crucial sensory characteristic of muscle foods, influences their functional acceptability. The smell perception can vary between a pleasant aroma and an unpleasant odour, directly related to the chemical nature of the volatile compounds formed by lipid peroxidation, their concentration, and the presence of non-lifeline components.

Major contributors to the formation of volatile compounds are lipoxygenases and the autoxidation of unsaturated fatty acids [388]. Reactive oxygen species are generated in muscle foods through enzymatic, chemical, photochemical pathways, or irradiation [83]. Hydroperoxides, formed during the initiation of lipid peroxidation, undergo breakdown into carbonyl derivatives, hydrocarbons, furans, etc., in subsequent stages of the process. Additionally, chemical interactions between reactive oxygen species and meat/fish ingredients lead to the creation of undesirable volatile compounds such as aldehydes, alcohols, carbonyls, and hydrocarbons [83], contributing to product deterioration.

The fragmentation of monohydroperoxides involves carbon–carbon cleavage on both sides of the alkyl radical chain, resulting in a range of volatile compounds, including alkenes, aldehydes, alcohols, esters, and carboxylic acids [14]. While some volatile compounds can be explained by this mechanism, others like furans, ketones, lactones, and aromatic products cannot [14]. The nature of volatile compounds formed depends on various factors, with the fatty acid profile of meat lipids being a crucial determinant influencing the number and proportions of hydroperoxide isomers [3,53]. Conditions under which hydroperoxides decompose strongly impact the composition of formed volatile compounds, along with oxidation mechanisms (autooxidation, thermooxidation, or photooxidation) and environmental conditions (temperature, pH value, presence of iron, etc.) [14,60].

Despite numerous volatile compounds forming during the oxidation of unsaturated fatty acids, key aromatic components include aldehydes, some unsaturated ketones, and furan derivatives [80,84]. These compounds exhibit a wide array of aromatic properties, with odours ranging from butter, oil, and fried oil to freshly cut grass, metal, cucumber, mushroom, and fruit. Volatile derivatives of n-3 PUFA have low perception thresholds, making them more influential in meat aroma and flavour compared to n-6 PUFA [138,140].

The breakdown of 13-linoleate (C 18:2) hydroperoxide results in the formation of hexanal and pentanal. The 9-hydroperoxide of the same fatty acid can produce methyl 9-oxononanate, 2,4-decadienal, and methyl octanoate. Hydroperoxides of linolenic acid (C 18:3) decompose into propanal, methyl octanoate, and 2, 4, 7-decatrienal [1,14,144]. Subsequent propagation of lipid peroxidation yields significant concentrations of mono- and dimeric hydroperoxides, decomposed by catalysis or acid thermal reaction. This process leads to the formation of volatile compounds, such as pentanal, hexanal, 2,3-octanediol, nonanal, and 2,4-decadienal in beef, worsening the meat flavour, and 1,5-octadienal-3-OH in cooked fish, with a sensing threshold of only 1.10^−12^ g.L^−1^ [147,152,153].

The 12-isozyme of lipoxygenase accelerates lipid peroxidation in minced grace carp (*Hypophthalmichthys molitrix*) compared to haemoglobin [53]. The resulting 2,4-heptadienal causes the appearance of a *warmed-over flavour* with a fishy smell, while hexanal and nonanal contribute to an oily smell. A strong correlation was observed between sensory characteristics, MDA content expressed by TBARS, and volatile lipid peroxide derivatives (hexanal and 2,3-octadione). Hexanal was suggested as an indicator of oxidative stability and, specifically, the acceptability of smoked pork flavour [170,196,377].

### 7.4. Negative Influence of Lipid Peroxidation on the Warmed-Over Flavour

In 1958, Tims and Watts were the first to coin the term “*warmed-over flavour*” (WOF) to characterise the rapid development of a rancid flavour in heat-treated meat during low-temperature storage. St. Angelo [386] attributed the off flavour in meat to the oxidation of membrane phospholipids and the significant formation of hexanal derived from linoleic acid. Palmitoleic (C 16:1), oleic (C 18:1), and linoleic (C 18:2) acids are identified as potential contributors to the formation of WOF [389]. Zhang et al. [389] established the specificity of the WOF phenomenon in different pig breeds, revealing its catalysis by metal complex ions, free radicals, and other generating systems.

St. Angelo [390] and Shahidi [391] reported several oxidation products associated with WOF, including propanal, butanal, pentanal, heptanal, 2,3-octadienal, and nonanal. Studies on WOF development during the storage of roasted meat shed light on the delicate balance between lipid oxidation products of phospholipids and Mallard reaction products influencing meat flavour [392]. The characteristic roasted meat aroma gradually transitions into an unpleasant WOF smell over time. Hypotheses suggest that phospholipids play a crucial role in WOF development in roast meat.

Contrary to Farmer’s hypothesis [391], which links WOF appearance to the masking of a pleasant aromatic-tasting bouquet due to increased content of unpleasant volatiles from lipid oxidation, a study on roasted turkey meat contradicts this notion. Phospholipid oxidation alone was not solely responsible for the off flavour in roasted turkey meat stored at 4 °C for four weeks. While the content of volatile oxidation products remained similar at the beginning and end of storage, significant changes were observed in Mallard reaction-derived compounds. The reduction in *Maillard* reaction products had a more significant impact on flavour changes in roasted meat than the presence of flavour compounds from lipid oxidation.

Three types of interactions contribute to the accumulation of similar compounds: oxidised lipid carbonyls and amino groups, amino groups of phosphatidylethanolamine and carbonyl groups of sugars, and free radicals from lipid peroxidation and Mallard reaction derivatives. Consequently, WOF in meat generates smells reminiscent of paint or cardboard, resulting in the loss of desired mild flavour notes, such as those of cooked beef or broth. The initially pleasant aroma of some sulphur-containing compounds during storage transforms into a rancid odour [386,393].

### 7.5. Lipid Peroxidation and Meat Taste

*Post mortem* changes in the lipid fraction of meat are closely linked to the development or deterioration of flavour, significantly influencing its shelf life [2,4,19]. Lipid degradation initiates immediately after death, involving two processes: hydrolytic degradation (lipolysis) and oxidative degradation (oxidation) [5,14]. Both processes contribute to the deterioration of meat taste [7,20]. Oxidative-type muscles demonstrate lower oxidative stability compared to glycogenic types [4,11]. Oxidised polyunsaturated fatty acids (PUFA) have been identified as responsible for the degradation of meat taste, with this effect intensifying during storage [386].

Red meat, poultry, and, to a lesser extent, fresh seafood possess a relatively mild taste on their own. However, after heat treatment, a specific meaty flavour develops [387]. The fishy flavour in fresh seafood is attributed to amines derived from trimethylamine oxide present in fish gadoids. Additionally, lipoxygenase-assisted oxidation leads to the generation of aldehydes and/or alcohols, particularly 2, 4, 7-dicatrienals and cis-4-heptenal, contributing to the seafood taste [387]. While non-volatile precursors of aromatic compounds play a role in the taste of cooked meat, amino acids, peptides, organic acids, nucleotides, etc., predominantly shape the meat taste [388]. Moreover, lipids play a key role in forming the appealing taste of heat-treated meat by participating in Mallard reactions with protein derivatives [2,4].

On the other hand, transition metal ions can initiate lipid peroxidation generating reactive oxygen species such as superoxyl anion radical (O_2_^−^·), peroxyl radicals (LOO·), and hydroxyl radical (HO·) [121]. The initiated lipid peroxidation is an autocatalytic process, and the resulting lipid hydroperoxides cross-link with proteins, deteriorating the palatability of muscle foods [126]. Protein oxidation induced directly by reactive species or indirectly through the reaction with secondary by-products of lipid peroxidation is critical for the sensory properties of muscle foods. For instance, cooking enhances free radical generation processes, suppresses the antioxidant defence systems of meat, and contributes to protein oxidation, leading to increased off-flavour [127].

### 7.6. Lipid Peroxidation and Meat Colour

The colour of fresh meat stands out as one of its most crucial characteristics, serving as a key factor by which consumers assess its freshness and quality. The attractiveness of fresh meat decreases as cherry red oxymyoglobin undergoes oxidation, resulting in the formation of grey–brown metmyoglobin [18,19]. In the presence of anions, at low pH values or high temperatures, oxymyoglobin loses oxygen and transforms into myoglobin. Subsequent oxidation, facilitated by the oxidised derivatives of lipid oxidation, leads to the formation of superoxyl anion radical (O_2_^−^·), hydrogen peroxide (H_2_O_2_) and metmyoglobin and ferric compounds of the type Fe^4+^ = O by myoglobin [13,18,20].

According to Shleikin and Medvedev [18] and Wu et al. [20], ferrous ions react with oxy-haem to produce met-haem. Muscle tissue also contains enzyme systems that reduce metmyoglobin back to oxymyoglobin, influencing the colour of meat and meat products. The mechanisms underlying oxidative changes in pale, soft, and exudative (PSE) chicken during storage in the dark at 4 °C for 5 days and after cooking at 80 °C for 30 min, as well as exposure to light and reheating, have been found. It was determined [92] that the oxidation of myoglobin, lipids, and proteins occurred both during the 5-day storage of PSE chicken meat in the dark at 4 °C and after cooking at 80 °C for 30 min.

Moreover, transition metals, especially metmyoglobin, are prone to generating free radicals that oxidise lipids and proteins. de Avila Souza et al. [92] suggest that light does not play a major role in lipid and pigment oxidation.

### 7.7. Lipid Peroxidation and Meat Texture

The interaction between peroxidised lipids and amino acids and proteins is a crucial aspect of lipid peroxidation in meat. This interaction occurs in three directions: the formation of covalent complexes, radical-type reactions producing covalent complexes, and reactions with secondary oxidation products [393,394]. According to Papuc et al. [11], the transfer of radicals in the protein polymerisation process occurs with the assistance of complexes between lipids and the sulfhydryl or M-canters of the reacting proteins. Radical reactions between oxidised lipids and proteins result in the formation of protein–protein or lipid–protein crosslinks and protein cleavage [390,395]. Secondary compounds such as aldehydes can react with amino groups and form Schiff bases [396].

A polymerisation effect takes place as a result of protein cross-linking. Polymeric proteins exhibit low solubility, reduced surface denaturation, and inhibited enzyme activity [392]. The interaction between lipids and proteins during the lipid peroxidation process significantly influences the texture properties of meat products, contributing to changes in their overall texture and consistency [7].

### 7.8. Lipid Peroxidation and Nutritional Value of Meat

Lipid peroxidation has a significant impact on the nutritional value of meat, as interactions with reactive oxygen species lead to the loss of essential nutrients and alterations in the functional properties of meat proteins and lipids [83]. This oxidative process is associated with the formation of various reactive molecular species, including hydroperoxides [138]. Derivatives resulting from lipid peroxidation contribute to negative changes, particularly in the autoxidation of unsaturated lipids, with fish, a primary source of PUFA, being notably affected, along with cholesterol. Extensive accumulation of toxic compounds has been identified [7,138]. Consequently, muscle lipid peroxidation is considered the mechanism behind the development of oxidative rancidity, representing a major factor in the gradual decline of the nutritional value of meat [144].

Moreover, accumulated free radicals have the potential to oxidise various vitamins, including carotenoids, and vitamins A, E, and C. Through subsequent processes, hydroperoxides may further oxidise many other molecules, notably cholesterol from membrane lipids. This dual action of lipid peroxidation results in a decrease in nutritional value and the formation of toxic substances [84]. The implications of lipid oxidation extend beyond the alteration of nutritional components, affecting overall meat quality and safety.

## 8. Effect of Oxidised Muscle Foods on Human Health

### 8.1. Introduction

The pathological condition characterised by the accumulation of free radicals and/or a diminished antioxidant defence is termed oxidative stress [83]. Cellular damage mediated by oxidative stress primarily results from the action of free radicals, which negatively affect lipids, proteins, and DNA, consequently contributing to various human diseases. In living organisms, a delicate balance exists between the processes involving free radicals and the protective antioxidant system. However, the known imbalance consistently favours oxidative processes. Lipids are stored and mobilised to different subcellular locations, where they can convey adaptive or maladaptive signals in myocytes [136]. According to Watt and Hoy [136], metabolic by-products of triacylglycerols or β-oxidation can function as both positive and negative regulators of insulin action.

Superoxyl anion radicals (O_2_^−^·) and reactive oxygen species play a dual role [128]. In the body’s physiological equilibrium, they act as by-products of decreasing oxygen levels required for cell signalling. Protein adducts also contribute to the deleterious effects of oxidative stress [81]. Considered harmful, they have the potential to induce pathological abnormalities such as apoptosis, necrosis, ferroptosis, pyroptosis, and autophagic cell death [128]. When there is a high rate of formation and a significant concentration of reactive free radicals, the antioxidant capacity of biological systems becomes insufficient to eliminate or inhibit their derivative products. Consequently, cell functions and structures undergo damage, toxic products of lipid peroxidation accumulate, and pathological processes ensue (Figure 8).

Reactive carbonyl species, whether in their free forms or as enzymatic or non-enzymatic conjugates resulting from lipid peroxidation, are considered indicators of oxidative stress in the human body [81]. Soladoye et al. [127] highlight the increasing interest among scientists in dietary approaches that mitigate the negative effects of lipid peroxidation products on human health and the ageing process. The focus is on the role of dietary lipids and their breakdown products in the development of conditions such as coronary heart disease (infarction), tumours, and stroke.

When oxidative changes occur, cells strive to maintain the necessary redox potential. However, if they cannot restore the functioning of damaged structures and sustain near-normal levels of degenerative processes, a disease state develops initially at the cellular and later at the systemic level. Certain organs and systems are more susceptible to damage from oxidative and nitrogen stress. These include the respiratory system (exposed to high concentrations of oxygen), the brain (with high metabolic activity but low levels of endogenous antioxidants), the eyes (constantly exposed to UV rays), the circulatory system (involved in transporting oxygen and nitrogen intermediates), and the reproductive system (with high metabolic activity in men) [127]. The impact of oxidative stress on these vulnerable organs and systems underscores the importance of addressing oxidative damage in the context of overall health and disease prevention.

### 8.2. Physiological Effects of Lipid Peroxidation Derivatives

The physiological impact of lipid peroxidation products from meat and fish manifests in various directions. Panov et al. [90] proposed the hypothesis that mitochondrial dysfunctions resulting from oxidative stress are associated with mutations and deletions in mitochondrial DNA (mtDNA), representing a hallmark of ageing. However, there is no evidence suggesting that most of the discussed oxygen radical species are directly responsible for mtDNA mutations [90]. In elderly individuals with metabolic syndrome, fatty acids become the primary substrates for adenosine triphosphate production, potentially leading to a several-fold increase in the generation of superoxyl anion radicals (O_2_^−^·) and, consequently, perhydroxyl radicals (HOO·). Metabolic syndrome has been demonstrated to accelerate ageing, with mitochondrial dysfunctions attributed to perhydroxyl radical-induced lipid peroxidation [133].

The dissociation of ferriprotoporphyrin IX (hemin) from globin and the release of iron atoms can contribute to various oxidative pathologies in muscle and muscle foods. This phenomenon is explained by the easy decomposition of lipid hydroperoxides into reactive oxygen species. Heme oxygenase and lipophilic free radicals can degrade the protoporphyrin IX moiety, resulting in the release of free iron. The presence of lipid hydroperoxides and low pH activates each of the oxidative forms of myoglobin, ferriprotoporphyrin IX, and the subsequent release of iron [133].

The reaction of free radicals of polyunsaturated fatty acids with molecular oxygen leads to the formation of lipid hydroperoxides [80]. Conjugated diene hydroperoxides at carbon atoms 9 and 13 are the primary products of bis-allyl 11-hydroperoxide of linoleic acid [95]. Yaman and Ayhanci [95] posit that malondialdehyde (MDA), the finished product of lipid peroxidation, is toxic as it causes fragmentation, modification, and aggregation of DNA and proteins (Figure 9). Excessive binding of such aldehydes to cellular proteins alters the permeability of cellular membranes and disrupts the electrolyte balance of the cell [8,10,24,76,82].

Protein adducts play a crucial role in cell signalling across various metabolic pathways. The formation of 4-hydroxy-2-nonenal protein adducts contributes to antioxidant responses by influencing transcriptional activity. Conversely, acrolein–protein adducts promote apoptosis, and protein adducts with polyunsaturated fatty acid (PUFA) cyclisation products primarily lead to inflammation or apoptosis [76].

Ferroptosis, on the other hand, is a genetically programmed, iron-dependent form of regulated cell death accelerated by enhanced lipid peroxidation and insufficient capacity of glutathione peroxidase 4 [106].

Opinions on the physiological action of hydroperoxides are conflicting. The lipid peroxidation chain reaction begins with the attack of free radicals on PUFA from cell membranes, forming hydrocarbon radicals (R). The superoxyl anion radical (O_2_^−^·) further attacks another lipid molecule, resulting in the formation of lipid hydroperoxide (ROOH). This chain reaction propagates and generates lipid perhydroxyl radicals (HOO·). Lipid hydroperoxides suppress the functionality of cell membranes and induce the diffusion of calcium ions through them. Severe damage to cell membrane phospholipids activates necrotic or apoptotic tissue death [95]. While hydroperoxides are not easily detected by the senses, in small concentrations, they inhibit the digestive enzyme systems in the stomach and intestines, disrupting nutrient absorption. At higher concentrations, hydroperoxides damage the epithelial tissue of the small intestine, become absorbed, and pass into the bloodstream. There, they degrade tocopherols and several other natural antioxidants and biologically active compounds [148]. The α, β-unsaturated hydroxyalkenal 4-hydroxynonenal (4-HNE) is produced by the lipid peroxidation of cell membranes. According to Ayala et al. [5], 4-HNE can either promote cell development or induce cell death (Figure 10).

At physiological levels, 4-hydroxy-2-nonenal (4-HNE) is enzymatically metabolised. At low concentrations, 4-HNE serves as a signalling molecule, stimulating gene expression, enhancing cellular antioxidant capacity, and triggering an adaptive response. At moderate concentrations within organelles, it causes protein damage and induces processes such as autophagy, senescence, or cell cycle arrest. However, at high or very high concentrations, 4-HNE promotes the formation of adducts and leads to apoptosis or cell death [5].

4-HNE is implicated in various pathophysiological conditions, playing a role in inflammation, regulating cell proliferation and growth, and contributing to necrotic or apoptotic cell death. It is discussed as a primary contributor to the pathogenesis of major chronic human diseases [205].

### 8.3. Autooxidation of Cholesterol and Human Health

Cholesterol in its pure form is not deemed atherogenic [397]. Atherosclerosis, however, is attributed to the presence of cholesterol oxidation products [398]. Numerous investigations into atherogenesis have demonstrated that the most atherogenic compounds arising from lipid oxidation are cholestane-triol and 25-hydroxycholesterol [399]. The absorption of cholesterol oxidation products occurs in trace amounts within the human body [400]. Substantial quantities of cholesterol oxidation derivatives are discernible in egg powder, heated fats or oils, and reheated meat products [401]. Approximately 2% of cholesterol in minced meat was identified as oxidised [402]. In ground reheated turkey meat, oxidised cholesterol products constitute about 3%, surpassing beef due to its faster oxidation rate [403].

There is compelling evidence suggesting that secondary products of lipid peroxidation in meat may impact human health [5,7]. In the process of lipid peroxidation, cholesterol molecules from the lipid fraction of cell membranes undergo oxidation. This intricate process yields numerous compounds capable of inducing atherosclerosis [398]. In the advanced stages of lipid peroxidation development, a number of carboxyl compounds accumulate as secondary products [13]. These compounds are readily absorbed in the intestines, leading to a substantial increase in their content in the lymph [398].

Research involving specific epoxides indicates that they do not exhibit pronounced toxicity but do possess carcinogenic effects [404]. Cholesterol oxides are acknowledged as atherogenic agents and likely demonstrate mutagenic, carcinogenic, and cytotoxic properties. They can substitute cholesterol molecules in membranes, disrupting their permeability, stability, and other inherent properties [397,399,400].

Common cholesterol oxidation substances found in food include 7-ketocholesterol, 7α-hydroxycholesterol, 7ß-hydroxycholesterol, 5,6-epoxides (5,6α-epoxycholesterol and 5,6ß-epoxycholesterol), triol—3ß-, 5α-, 6ß-trihydroxycholesterol (also known as cholestane-triol), as well as 25-hydroxycholesterol and 20α-hydroxycholesterol [399,400]. Notably cytotoxic among these are cholestane-triol and 25-hydroxycholesterol [397], with the latter inhibiting the activity of 3-OH-3-CH_3_-glutaryl coenzyme A reductase (HMG-CoA reductase) and retarding the synthesis of endogenous cholesterol [405].

Oxysterols from food can be absorbed by the human body [406], transitioning briefly through chylomicrons in the metabolism before swiftly transforming into other plasma substances, primarily other lipoprotein fractions [407]. The oxidation of cholesterol occurs more expeditiously in foods prepared through drying and/or prolonged storage, such as freeze-dried pork [408,409]. This accelerated oxidation contributes to the generation of a variety of chemical compounds capable of triggering atherosclerosis [397].

### 8.4. Coronary Cardiovascular Diseases

Oxidative stress is intricately linked with coronary heart disease [410]. The statistics for 2021 underscore the gravity of this issue, with approximately 695,000 individuals succumbing to it, constituting one in five deaths [411,412].

The type and quantity of ingested lipids have been identified as pivotal predictors of atherosclerosis [397]. Plaques or atheromas develop in the intima of arterial walls due to accumulations of cholesterol and lipids [413]. Atheromas form through blood lipid–protein complexes known as low-density lipoproteins (LDL) [414]. It is probable that the LDL fraction escalates the risk of coronary heart disease [77]. The cardiotoxicity associated with cholesterol oxidation products significantly contributes to the onset of secondary cardiovascular diseases. Cholesterol oxides generate deposits in the heart and aorta, leading to plaque formation in the circulatory system. As a potential risk factor for atherosclerosis, Caslake et al. [77] implicated lipoprotein phospholipase A2 with platelet-activating factor acetyl hydrolase. According to these researchers, lipoprotein phospholipase A2 plays a role in the oxidative modification of LDL cholesterol [415]. Very low-density lipoprotein (VLDL) cholesterol does not appear to be atherogenic and does not induce atherosclerosis. High-density lipoproteins (HDL) transport cholesterol away from the circulatory system and act as inhibitors of plaque formation [416].

The structural and functional damage resulting from cardiac muscle reperfusion in operative treatment is as severe as the damage causing ischemic heart disease [417]. This is likely attributed to oxidative stress arising from increased formation of reactive oxygen species and secondary radicals [418]. Frequently, the hypoxic state is succeeded by subsequent deoxygenation, stemming from the restoration of blood flow or the treatment of ischemic tissue under aerobic conditions. This invariably leads to pathological changes in the tissues, a phenomenon recognised as the oxygen paradox [419].

### 8.5. Stroke

There is compelling evidence [420] suggesting that the modification of plasma LDL is responsible for the pathogenesis of atherosclerosis. Oxidatively modified LDL and their derivatives have been identified as key contributors to a significant array of pro-inflammatory and pro-atherogenic properties. Hypotheses supporting the pivotal role of specific reactive aldehydes, products of oxidised LDL, have emerged as crucial signalling molecules associated with atherosclerotic lesions. These aldehydes exhibit facile reactivity towards proteins, forming intra- and intermolecular covalent adducts on apo-lipoprotein B-100 in LDL [421].

Due to established specific regional vulnerability to lipid peroxidation, hundreds of millions of people in different parts of the world are believed to be at risk of high blood pressure and an increased likelihood of developing stroke, myocardial infarction, and kidney disease [154]. Naudí et al. [154] reported distinct neural mechanisms of PUFA biosynthesis in the human central nervous system. Stroke tends to predominate in obese individuals as high-fat diets contribute to its development, and atherosclerosis partially obstructs the arterial system, leading to elevated blood pressure [422].

Recently, more attention has been directed towards the more prevalent type of stroke—ischemic. It has been established that an elevated level of cholesterol lipoprotein, commonly known as “bad cholesterol,” poses a particular risk to blood vessels. Excessive levels in the blood lead to the formation of atherosclerotic deposits, or plaques, accumulating on vessel walls and impeding the free flow of blood. Consequently, the organ supplied with blood (heart or brain) receives diminishing oxygen. Atherosclerotic plaques can eventually detach, forming a thrombus that completely blocks the vessel, resulting in the death of part of the brain tissue and triggering an ischemic stroke. This type of stroke is more prevalent, accounting for 80–85% of all strokes. While the elderly are most susceptible, middle-aged and even younger individuals are not exempt [419]. The high incidence of strokes is attributed to ineffective prevention, an unhealthy lifestyle, and the neglect of risk factors, with high blood pressure being the foremost concern, not exceeding 120/80. Patients are advised to continuously monitor their blood pressure [420].

### 8.6. Neurodegenerative Diseases and Oxidative Stress

Oxidative stress has been demonstrated to play a pivotal role in the development of neurodegenerative diseases [423]. Individuals afflicted with neurodegenerative diseases undergo a progressive loss of memory and cognitive abilities, often accompanied by changes in mood, personality, and a sense of diminished independence. In this context, Mohammed and Ibrahim [424] consider various free radical reactions, triggered by the formation of an initiating radical. Three potential mechanisms are proposed for this activation: through a redox cycle, via a thermal process, or through a photochemical process. Consequently, the oxidation of nucleic acids, proteins, and lipids is sustained, leading to the accumulation of health-detrimental oxidised derivatives.

Das et al. [100] delve into the pathophysiology of Alzheimer’s disease, focusing on the neuroinflammatory and neurodegenerative effects of transition metals. They posit that transition metals involved in Fenton reactions are accountable for neuronal plasticity and neuroprotection. It has been hypothesised that reactive oxygen and nitrogen species exert pathological effects, contributing to various neurological disorders [100]. Mohammed and Ibrahim [424] and Polidori et al. [423] also propose a hypothesis concerning the role of selective oxidative stress on the central nervous system. According to Mohammed and Ibrahim [424], xanthine oxidase is responsible for generating reactive oxygen species during the deoxygenation of ischemic tissues. They contend that ageing, inflammatory processes, viral infections, and neurodegenerative diseases depend on the actions of reactive oxygen species, which can escalate under the influence of various growth factors and cytokines associated with different types of receptors on the cell membrane. Beyond Alzheimer’s disease, Mohammed and Ibrahim [424] explore the etiopathologies of Parkinson’s disease and Huntington’s chorea.

### 8.7. Influenza Virus Infection and Oxidative Stress

Studies conducted by Chen et al. [425] and Darenskaya et al. [426] have substantiated a direct association between the development of influenza virus infection and oxidative stress, extending beyond the lungs to impact other organs and systems. The sequential order in which individual organs are affected, along with the intensity of viral impact, follows the upper respiratory tract and lungs, liver, heart, and brain. Infection leads to lung inflammation, disrupting not only respiratory function but also the organ’s detoxification capacity and general cell permeability. Oxidative stress plays a crucial role in both the pathogenesis and infectivity of the influenza virus. It has been demonstrated that mortality in influenza patients results not from the direct effects of the virus but from the deleterious impact of free radical mediators and reactive oxygen species induced during and in response to the infection [427].

Upon entering the body, the virus encounters antigen-presenting cells that release highly reactive metabolites, active forms of oxygen [428].

### 8.8. Rheumatoid Arthritis and Oxidative Stress

Rheumatoid arthritis is a chronic systemic disease characterised by an autoimmune component [107]. It involves humoral and cellular mechanisms associated with the formation of immune complexes, which are responsible for the production of oxygen radicals. These radicals, in turn, inflict damage on membranes and activate phospholipase A2. This activation contributes to the release of arachidonate from membrane phospholipids, leading to an increased accumulation of prostanoids. The presence of oxygen radicals in the system results in the complete deactivation of microsomal cytochromes, accompanied by the impairment of corticosteroid biosynthesis. Consequently, symptoms of anaemia manifest, and there is an accumulation of inorganic iron [107].

Sutipornpalangkul et al. [429] posit that free radicals play a significant role in the pathogenesis of knee osteoarthritis. They attribute the damage to proteins, lipids, nucleic acids, and matrix components to the excessive production of reactive oxygen species, products of abnormal chondrocyte metabolism. According to these researchers, reactive oxygen species surpass the physiological buffering capacity, causing oxidative stress. Moreover, they serve as intracellular signalling molecules that amplify the inflammatory response.

### 8.9. Kidney Diseases and Oxidative Stress

Nephrological diseases encompass a group of pathologies arising from disorders in the excretory and filtering functions of the kidneys [430]. Common clinical–functional manifestations associated with kidney diseases include hypertension, oedema, anaemia, various homeostatic and haemostatic disorders, osteodystrophy, alterations in diuresis and urine composition, as well as a diminished or lost concentration dilution capacity [431]. It is well-established that oxidative stress and free radical peroxidation of lipids form the basis of the mechanisms driving these pathological changes. The heightened formation of reactive oxygen species, generated by activated neutrophils, macrophages, and mesangial cells in the kidneys, has been demonstrated to induce cellular damage. Consequently, oxidative stress disrupts essential structural and functional properties of biological membranes, including fluidity, ion transport, enzyme activity, and a decrease in the overall oxygen content in the body [432].

### 8.10. Liver Disease and Oxidative Stress

Webb and Twedt [433] have identified an association between oxidative stress and liver pathophysiology. In light of this, the discussion explores the potential treatment avenues for patients with liver diseases through the use of suitable antioxidants.

### 8.11. Disorders in Erythropoiesis, Leukaemia and Oxidative Stress

Anaemia is a condition characterised by a single and/or combined decrease in the number of erythrocytes and haemoglobin in a unit volume of blood. Given that erythrocytes play a crucial role as carriers of oxygen and carbon dioxide, anaemias can be considered a specific manifestation of hypoxia. Various criteria, such as the volume and haemoglobin concentration of the average erythrocyte, aetiology, pathogenetic principle, and clinical course, are employed to classify anaemias [434].

In a study by Rác et al. [435], cell suspensions from the human leukemic monocytic lymphoma cell line U937 were investigated. The extraction of hydrogen peroxide and Fenton’s reagent led to the oxidation of lipids, proteins, and DNA. The addition of hydrogen peroxide to the cell suspension, as per Rác et al. [435], resulted in the formation of a hydroxyl radical through an endogenous metal-mediated Fenton reaction. Oxidative damage to biomolecules occurs through the decomposition of high-energy intermediates like dioxetane or tetroxide, forming singlet oxygen. In contrast to enzymatic degradation, reactive oxygen species randomly attacked all carbon bonds of the tetrapyrrole rings, generating various pyrrole products and releasing free iron [116]. According to Nagababu and Rifkind [116], only non-enzymatic degradation of haem occurs in red blood cells. Heme iron undergoes redox cycling reactions in the presence of reactive oxygen species.

Thalassemia, often referred to as Mediterranean anaemia due to its prevalence in that region, is a group of hereditary acute diseases that typically manifest in infancy or childhood. The disease is characterised by metabolic imbalance, the accumulation of free iron in the body, chronic hypoxia, and cell damage. Pathophysiological changes in thalassemia result from ineffective erythropoiesis, haemolysis, and subsequent anaemia [436].

Homozygous β-thalassemia major is clinically marked by severe anaemia and organ changes associated with chronic hypoxia and elevated serum iron [437]. Iron overload initiates a cascade of disease processes involving oxidative damage to the erythrocyte membrane and other cellular membrane structures. This involves metabolic hyperproduction of reactive oxygen species and free radicals, activation of free radical peroxidation of lipids, and the formation of lipid peroxidation products [438]. The increased accumulation of radical intermediates results in an imbalance in the body’s antioxidant capacity (including endogenous antioxidants and antioxidant enzymes). These changes underscore oxidative stress as a leading mechanism in the development of β-thalassemia major [439].

### 8.12. Carcinogenesis

Devasagayam et al. [440] proposed that lipid peroxidation leads to protein damage and the loss of enzyme activity. Subsequently, damage to DNA can result in mutagenesis and carcinogenesis [440]. Reactive oxygen species play a significant role in the formation of carcinogens and the cross-linking or cleavage of proteins in muscle foods, altering the functionality of proteins, lipids, and carbohydrates [83]. At high concentrations, reactive oxygen species damage biomolecules in cell membranes and are considered responsible for the aetiology of various diseases, including atherosclerosis, diabetes, neurodegenerative diseases, chronic inflammation, cancer, and ageing. The hydroxyl radical, for instance, can damage DNA and induce lipid peroxidation and protein modification [113]. Additionally, variable valence metal-mediated free radical formation can cause varying degrees of DNA modification, enhanced lipid peroxidation, and changes in calcium and sulfhydryl homeostasis [192]. According to Valko et al. [192], lipid hydroperoxides formed by radical attacks on polyunsaturated fatty acid residues of phospholipids can further react with redox metals to produce mutagenic and carcinogenic malondialdehyde, 4-HNE, and other exocyclic DNA adducts. Valko et al. [195] assert that there is sufficient evidence to suggest that reactive oxygen species induce and maintain the oncogenic phenotype of cancer cells through a complex intracellular mechanism. Oxidative stress induced by reactive oxygen species can also lead to cellular senescence and apoptosis. The authors argue that oxidative stress induces a cellular redox imbalance, a phenomenon found in various cancer cells in contrast to normal ones. This redox imbalance is associated with oncogenic stimulation, and DNA mutation is a critical step in carcinogenesis, with increased levels of oxidative DNA lesions (8-OH-G) found in various tumours. Valko et al. [195] describe the mechanisms of carcinogenesis and the role of reactive oxygen species in signalling cascades, pointing particularly to the reactive oxygen species-activated activator protein (AP-1) and kappa B (NF-κB) nuclear factor signal transduction pathways. These pathways can cause the transcription of genes involved in the regulatory pathways of cell growth.

Cancer is considered one of the most common causes of mortality worldwide [441]. Breast and uterine cancer have been associated with obesity in women [442]. PUFA may increase the risk of cancer due to their susceptibility to autoxidation [192]. Studies in experimental animals have indicated that tumour incidence is reduced when calories, PUFA, and protein are reduced in the diet [443]. Other compounds in dietary lipids responsible for carcinogenesis include MDA, cholesterol oxides, and possibly other oxidation products [432].

A high fat intake (40–45% of the energy value of food) has been linked to an increased risk of breast, colon, pancreatic, prostate, and lung cancer [444]. The stimulating effect of fats is associated with both the high intake of saturated fatty acids, primarily from animal fats, and the excessive intake of PUFA, which is prevalent in the diet through sources like sunflower oil [445].

Excessive intake of n-6 PUFA, mainly obtained from vegetable fats, may stimulate the development of tumours induced by carcinogenic chemical substances entering the body [446]. However, n-3 PUFA, characteristic of fish oils, inhibit the action of chemical carcinogens [447].

The likely mechanism of the fat’s action depends on the location of the tumour. For example, in the case of colon cancer, the stimulating action of fats is thought to be due to the release of bile acids, which, in turn, stimulate the development of cancer processes. In breast cancer, the effect of fat is believed to result from fat-induced changes in the sex hormone profile [448].

## 9. Conclusions

The review of the available literature leads to the conclusion that the processes of lipid peroxidation in muscle foods have been extensively studied, primarily from a theoretical perspective and less so from an applied standpoint. While numerous publications exist, a comprehensive systematic approach has not always been employed in studying lipid peroxidation concerning the processing and storage of meat and fish products, as well as their impact on quality and nutritional value. This highlights the relevance of further research in this area.

Investigating lipid oxidation in relation to muscle metabolism can offer valuable insights into the formation of volatile molecules in heat-processed meat and fish products, cured meat, and dry-salted fish products. The findings may be instrumental in controlling and optimising technological processes.

The impact of natural antioxidants, either inherent in muscles through dietary intake or introduced as additives during technological processing, has not been fully and thoroughly examined. The interactions between sensory properties, the formation and accumulation of toxic compounds, lipid oxidation pathways, and strategies to prevent or inhibit these negative processes for the quality and safety of muscle food present an intriguing area for further study. The oxidative stability of fresh meat and fish, as well as heat-treated and dry-fermented meat and fish products, poses a challenge as it decreases during storage, affecting all participants in the food chain. Hence, one of the main objectives of meat science is to establish the relationship between lipid peroxidation in muscle foods and human health. Future research should explore the potential emergence of toxic molecules resulting from lipid oxidation. Further studies are needed to formulate appropriate technological solutions to reduce the risk of chemical hazards caused by the lipid peroxidation processes in muscle foods.

## Figures and Tables

**Figure 1 foods-13-00797-f001:**
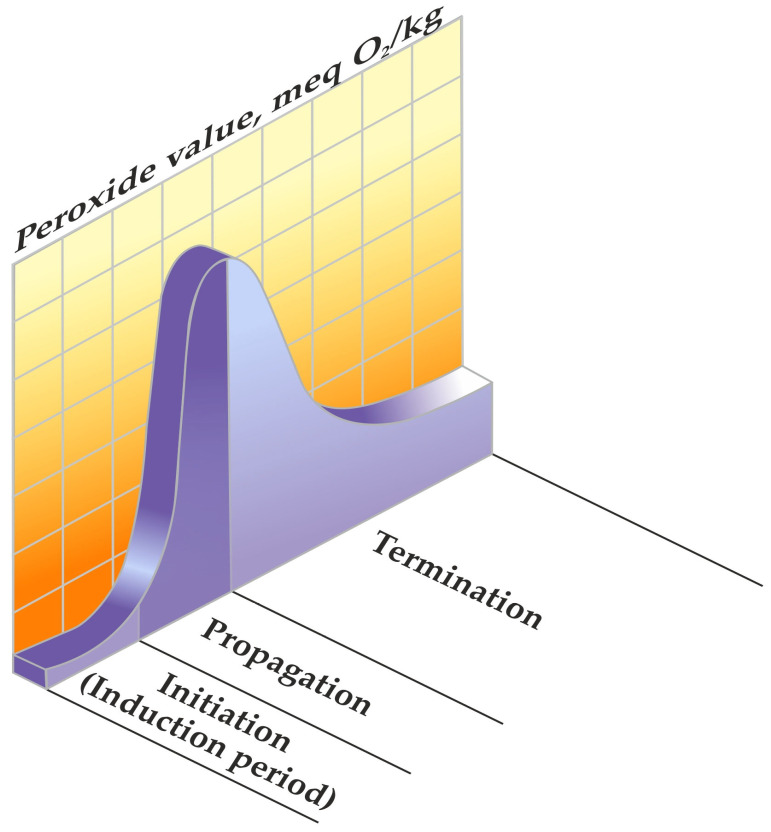
Lipid peroxidation periods: formation of hydroperoxides and their conversion into secondary products responsible for the unpleasant flavour of muscle foods.

**Figure 2 foods-13-00797-f002:**
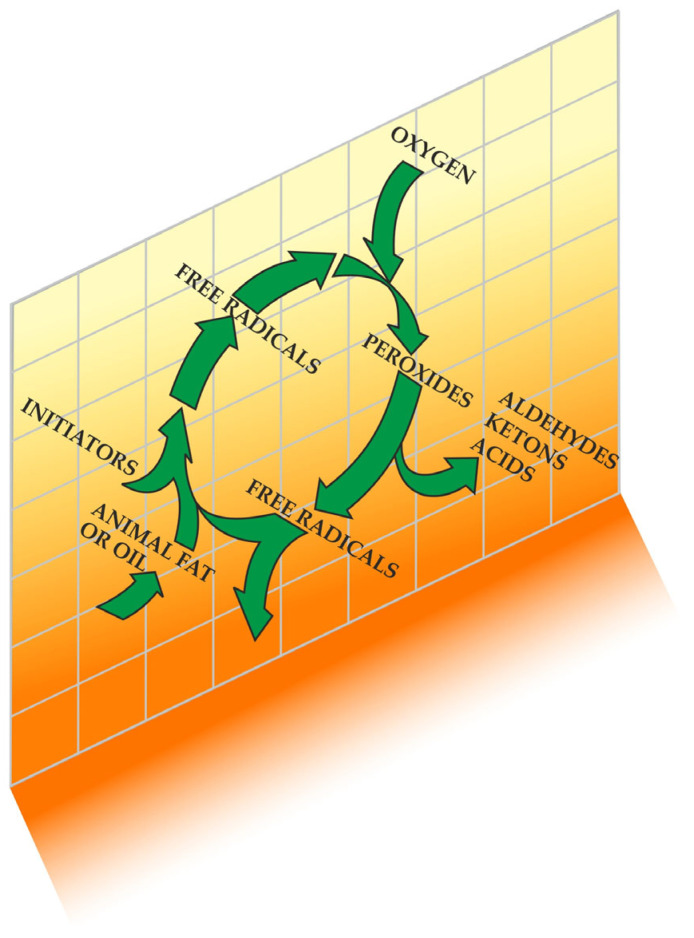
Lipid peroxidation cycle: the cause of progressive autooxidation of lipids and pigments, leading to the deterioration of sensory quality and nutritional value of muscle foods.

**Figure 3 foods-13-00797-f003:**
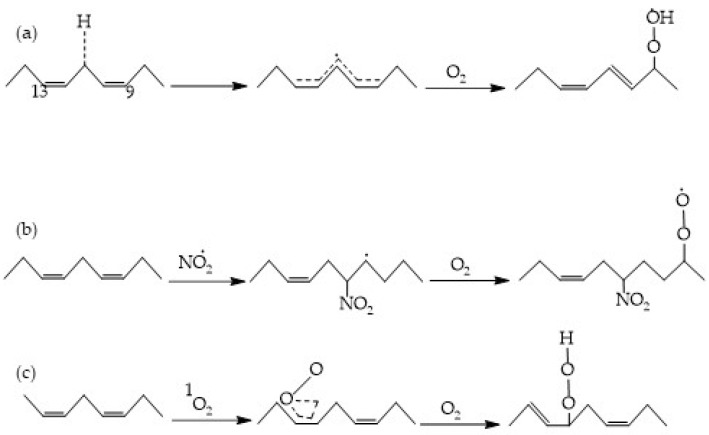
Initiation of lipid peroxidation: (**a**) by separation of hydrogen; (**b**) by free radicals attacking a double bond; (**c**) by singlet oxygen in the so-called “en” reaction (adapted by Kanner [7] Copyright year 1994, Elsevier).

**Figure 4 foods-13-00797-f004:**
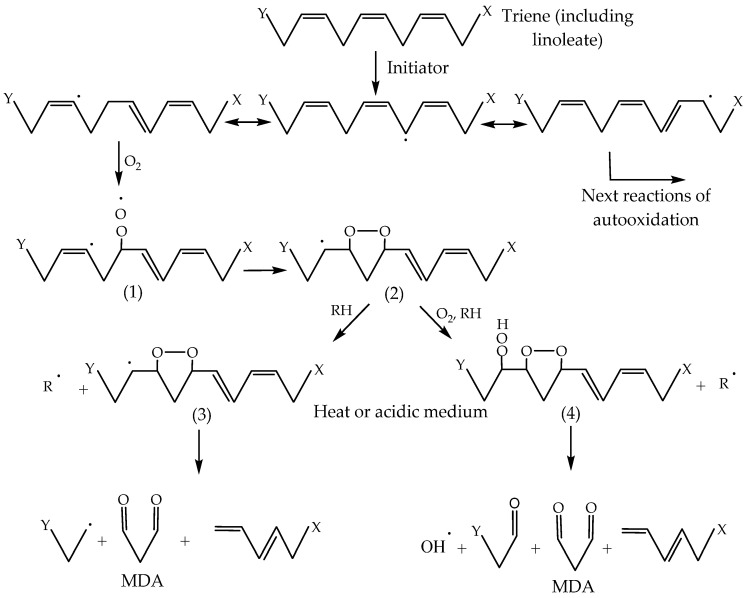
The malondialdehyde formation in a triene system (reprinted with permission from Pryor et al. [79] Copyright year 1976, AOCS).

**Figure 5 foods-13-00797-f005:**
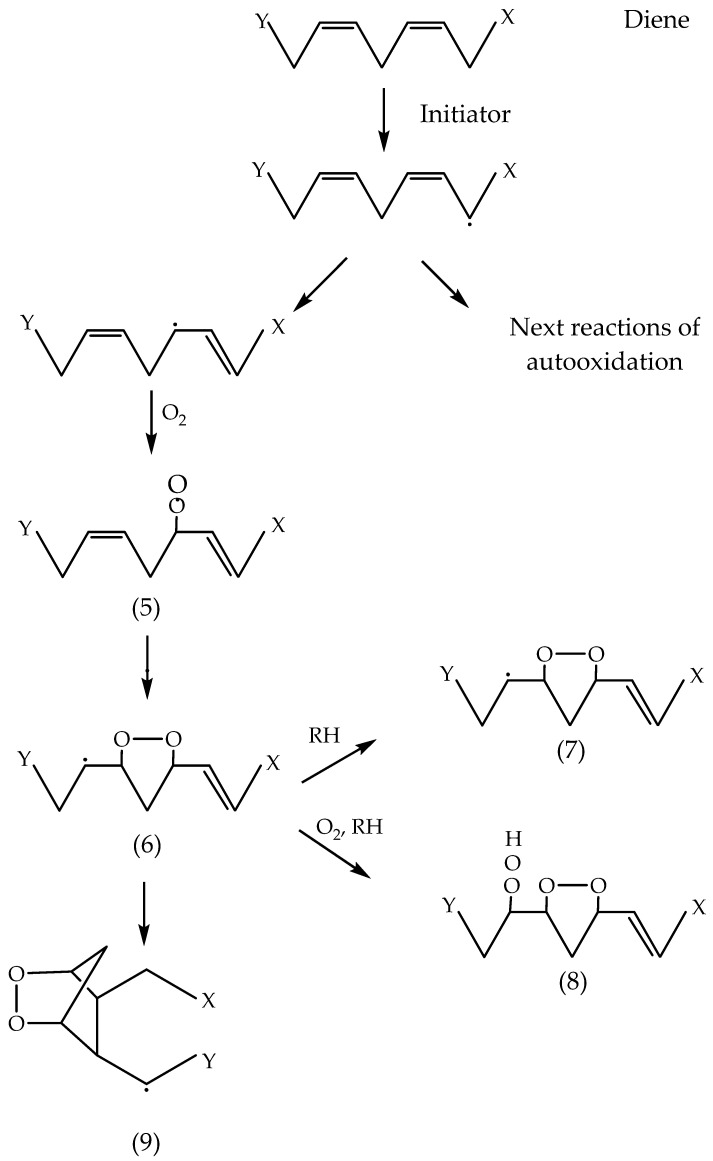
The malondialdehyde (MDA) precursor’s formation in a diene system (reprinted with permission from Pryor et al. [79] Copyright year 1976, AOCS).

**Figure 6 foods-13-00797-f006:**
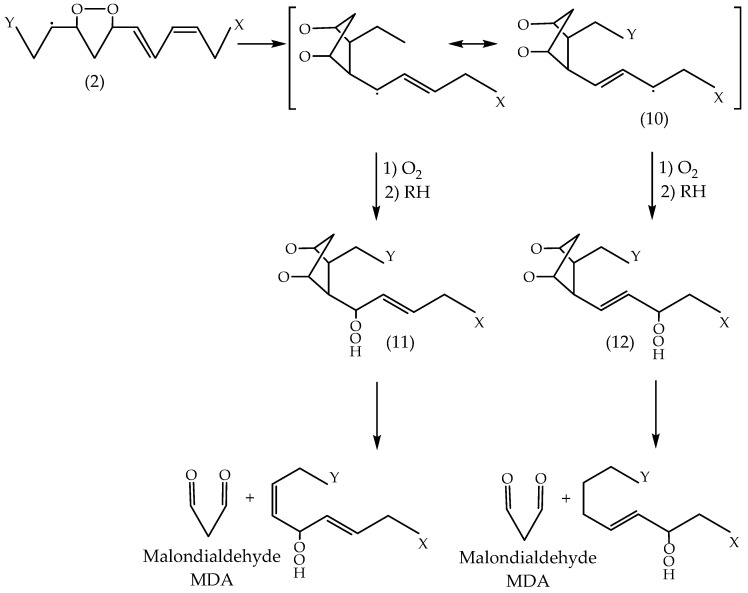
Prostaglandin type of endoperoxide mechanism for the malondialdehyde (MDA) formation in a diene system (reprinted with permission from Pryor et al. [79] Copyright year 1976, AOCS).

**Figure 7 foods-13-00797-f007:**
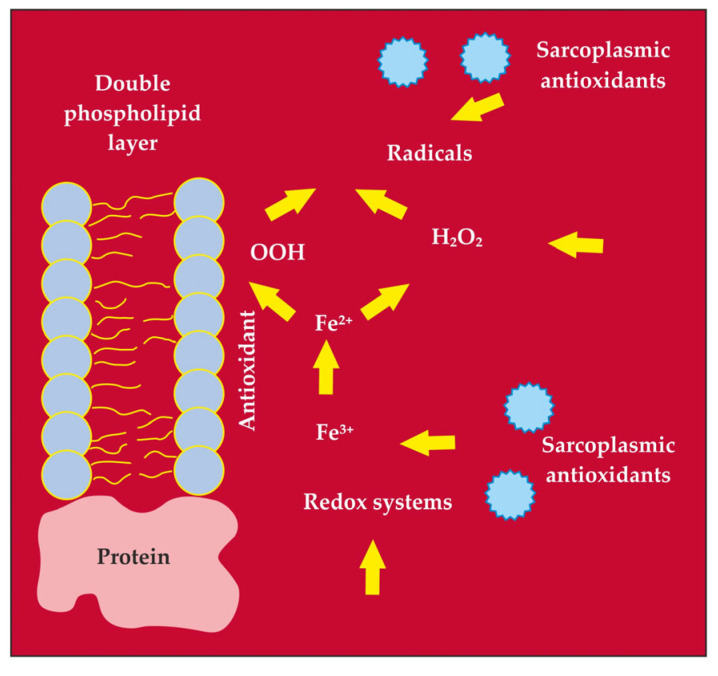
Endogenous constituents of the skeletal muscles included in the compensatory mechanisms and systems that maintain the balance between the factors controlling lipid peroxidation (adapted by Decker and Xu [135] Copyright 1998, authors/Institute of Food Technologists).

**Figure 8 foods-13-00797-f008:**
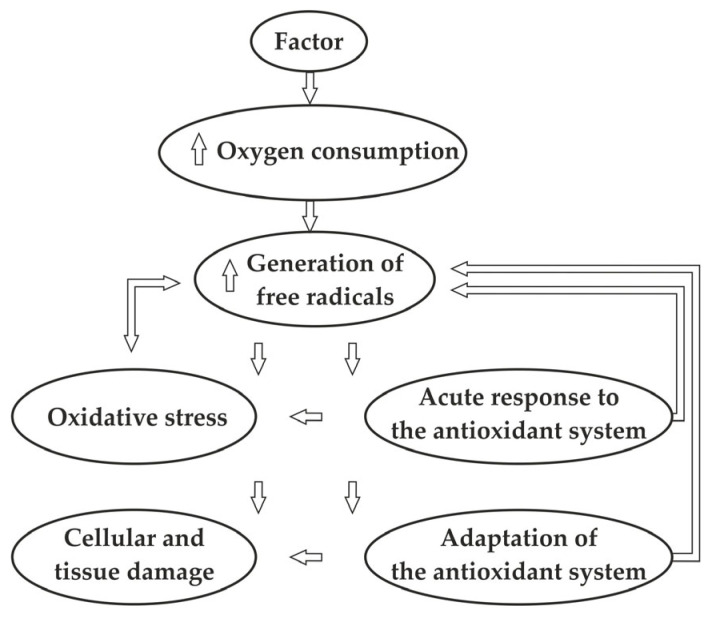
Flow diagram of oxidative stress induction in living biological systems.

**Figure 9 foods-13-00797-f009:**
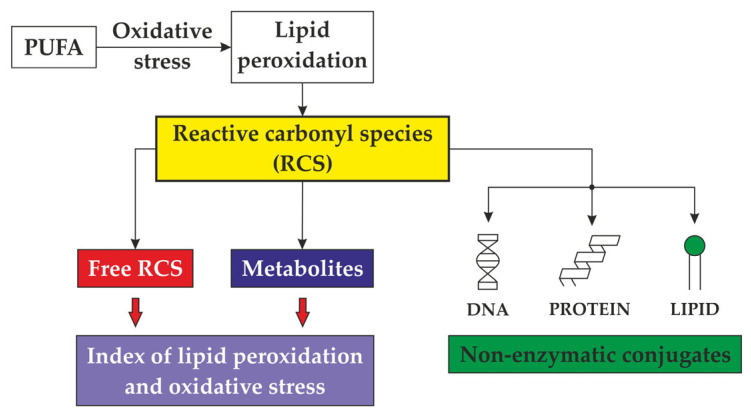
Mechanism of formation of free forms or enzymatic and non-enzymatic conjugates of reactive carbonyl species formed during lipid peroxidation, used as an index of oxidative stress. (adapted from Altomare et al. [81] Copyright year 2021, Elsevier).

**Figure 10 foods-13-00797-f010:**
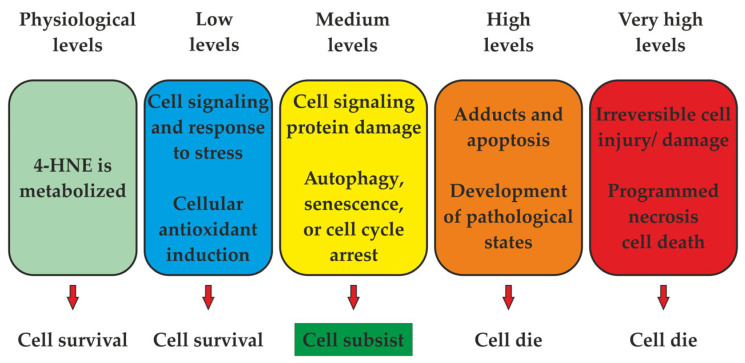
4-Hydroxynonenal (4-HNE) promotes cell survival or induces cell death adapted with permission from Ayala et al. [5].

## Data Availability

The original contributions presented in the study are included in the article, further inquiries can be directed to the corresponding author.

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
