# Peer review of "Lipid Peroxidation in Muscle Foods: Impact on Quality, Safety and Human Health"

_foods, 2024, doi:10.3390/foods13050797_

Round 1
Reviewer 1 Report
Comments and Suggestions for Authors
The current review by Stefan G. Dragoev delves into a captivating subject, thoroughly examining the literature on Lipid Peroxidation in Muscle Foods: Impact on Quality, Safety, and Human Health.
Title
While the title is generally effective and reflective of the paper's findings, the term "muscle foods" may benefit from a slight modification for improved clarity.
Abstract:
In line 21, the authors aptly describe their work as a "critical review," but there is a subtle difference in line 26, where it is simply termed a "review." Clarification between a review paper and a critical review paper is needed. Additionally, lines 26-28 require refining for greater coherence.
Introduction:
In lines 34-36, an essential addition would be a reference to support the information presented. Attention to this detail enhances the scholarly foundation of the manuscript.
Line 82 should introduce "Figures" as Figure 1, Figure 2, and so forth, for clarity instead of Fig 1, 2,3. Additionally, commonly used abbreviations such as NADPH, ADF, and ADP need to be spelled out when first introduced, fostering a better understanding for readers.
Concerning the section on the Enzymatic Initiation of Lipid Peroxidation in Meat Products (line 121), there is a recurring ambiguity about the type of meat products. To rectify this, the term "Meat Muscles" is suggested for precision.
Pages 8, 9, and 10 contain valuable information well-explained for readers. Commendations for the clarity and usefulness of this section.
Pages 10 to 15 raise a concern about an imbalance of chemistry-related content. It is recommended to provide more information on actual lipid peroxidation in meat muscles to maintain a well-rounded perspective. This note applies to subsequent pages up to page 23, where there appears to be an overemphasis on chemistry.
Line 493, Point 5.6, addressing Singlet Oxygen, requires an explanatory legend for the accompanying figure.
Line 583 suggests replacing "Introduction" with "Background" as a subheading in the main body of the paper for better alignment with scholarly conventions.
Line 867 acknowledges rosemary's antioxidant potential but suggests expanding the discussion to include other proven plant extracts like for example grape pomace, basil, sage, oregano, thyme, and artemisia.
Line 939 recommends placing references appropriately rather than combining them into one reference.
Line 941. Why only the technological methods are discussed? It will be interesting to mention briefly other methods used for inhibition of lipid peroxidation in meat muscles. I think that some of these details can be found here (https://doi.org/10.3390/foods12214001), based on the findings from the literature.
Other methods besides the technological ones could be:
Enzyme Inhibitors involved in lipid peroxidation, such as lipoxygenase inhibitors. These can help disrupt the enzymatic initiation of peroxidation.
Irradiation reduces microbial load, extending the shelf life of muscle foods and indirectly inhibiting lipid peroxidation.
Heat treatment during processing to inactivate enzymes and reduce the likelihood of lipid peroxidation.
Microbial fermentation can produce bioactive compounds and metabolites that act as antioxidants and inhibit lipid peroxidation.
Another observation is that this part is mostly focused on fish meat, and the rest of it (chicken, pork, and beef is neglected). Addressing the neglect of chicken, pork, and beef in this section will enhance it.
Line 1303 needs review to ensure it conveys the intended message clearly.
Line 1247, titled "Effect of Oxidized Muscle Foods on Human Health," could benefit from a more focused exploration of the actual health effects, potentially incorporating examples from clinical studies.
Line 1593's final sentence should be rephrased as a recommendation for further research.
Pros:
The paper is compelling, of high quality, and warrants consideration for publication in the Foods Journal.
Cons:
While the paper is well-researched, the extensive focus on chemistry and reactions, particularly those already studied, may require adjustment. There's also a noted discrepancy between the first part, covering Lipid Peroxidation, and the second part, addressing the Impact on Quality, Safety, and Human Health.
The author is encouraged to consider addressing these concerns for an even more refined manuscript.
Good luck!
Author Response
REVIEWER 1
Comments and Suggestions for Authors
The current review by Stefan G. Dragoev delves into a captivating subject, thoroughly examining the literature on Lipid Peroxidation in Muscle Foods: Impact on Quality, Safety, and Human Health.
Thank you very much for taking the time to review this manuscript. Thank you also for the assessment on the thoroughness of the topic and the literature review of the article. My team and I have been working on this topic for over 20 years and have dozens of publications in this field. The topic is important for science in general because it directly affects the safety and quality of meat and fish products. That's why I've tried to include as many publications as possible which present to the reader as comprehensive a view of the problem.
Please find detailed responses below and corresponding revisions/corrections, highlighted/tracked changes in the resubmitted files. I am responding to your suggestions point by point:
Title
While the title is generally effective and reflective of the paper's findings, the term "muscle foods" may benefit from a slight modification for improved clarity.
Thank you so much for the "flattering" assessment that the title is generally effective and reflective of the paper's findings. The term "muscle food" is often used in the meat science literature. For example, I present to your attention links to some similar articles from the last few years:
https://doi.org/10.1016/j.meatsci.2016.09.017
https://doi.org/10.1111/1541-4337.13040
https://doi.org/10.1016/j.foodcont.2017.06.009
https://doi.org/10.3390/antiox11010060
https://doi.org/10.1111/1541-4337.13064
https://doi.org/10.1111/1541-4337.12604
Initially, it was discussed that the first part of the title should be: "Lipid Peroxidation in Meat and Fish". As you have noticed, in the text of the article, not only fresh red and white meat or fish are considered, but also a number of meat and fish products with a variety of production technology. If in some sections the term "meat and fish" would fit perfectly, then in other cases it was necessary to use meat products, or fish products or poultry products, etc. This is the reason why I also looked for a more general term for this very diverse a group of foods of animal origin. The unifying feature was that they were all produced from a raw material mainly made up of muscle tissue. For this reason, I refrain from correcting the term "muscle foods" both in the title and further in the text.
Abstract:
In line 21, the authors aptly describe their work as a "critical review," but there is a subtle difference in line 26, where it is simply termed a "review." Clarification between a review paper and a critical review paper is needed. Additionally, lines 26-28 require refining for greater coherence.
Thank you very much for your opinion, and indeed there is a subtle difference between "critical review" and just "review" in line 26. The inconsistency has been fixed by using the term "article" on line 26.
Additionally, the sentence in lines 26-28 has been refined for greater coherence. The last sentence of the abstract was deleted and a new, more clearly worded one was put in its place:
The review concludes by emphasizing the need for the scientific community to focus on identifying appropriate technological solutions to reduce the risk of lipid oxidation in meat and fish products. Further studies are also needed to formulate appropriate technological solutions to reduce the risk of chemical hazards caused by the initiation and development of lipid peroxidation processes in muscle foods.
Introduction:
In lines 34-36, an essential addition would be a reference to support the information presented. Attention to this detail enhances the scholarly foundation of the manuscript.
Thanks to dear Reviewer 1 for the correct note about a reference to support the information presented in lines 34-36. Unfortunately, I cannot make a reference, because the text from lines 36 - 37 is my summaries and reflections on the discussed topic. As you notice the next sentence - lines 37 - 38 is correctly accompanied by a citation of a source [1] that was used.
Line 82 should introduce "Figures" as Figure 1, Figure 2, and so forth, for clarity instead of Fig 1, 2,3.
Thank you so much for pointing this out. I have corrected it.
Additionally, commonly used abbreviations such as NADPH, ADF, and ADP need to be spelled out when first introduced, fostering a better understanding for readers.
You are right, and I accept the Reviewer1’s opinion that commonly used abbreviations such as NADPH, ADF, and ADP need to be spelled out when first introduced, fostering a better understanding for readers. I have corrected them independently the list of used abbreviations is presented (see lines 1603 – 1627 of the initial version of the manuscript).
Concerning the section on the Enzymatic Initiation of Lipid Peroxidation in Meat Products (line 121), there is a recurring ambiguity about the type of meat products. To rectify this, the term "Meat Muscles" is suggested for precision.
Thank you very much for this remark. For clarification and accuracy, the term "meat products" has been replaced with "red meat" - the subject of discussion in this sentence.
Pages 8, 9, and 10 contain valuable information well-explained for readers. Commendations for the clarity and usefulness of this section.
I express my immense gratitude to Reviewer 1 for the flattering assessment of the scientific value of the text on pages 8-10.
Pages 10 to 15 raise a concern about an imbalance of chemistry-related content. It is recommended to provide more information on actual lipid peroxidation in meat muscles to maintain a well-rounded perspective. This note applies to subsequent pages up to page 23, where there appears to be an overemphasis on chemistry.
Thank you so much for this question and for your valued opinion and suggestions made for improvements to the part of manuscript in pages 10 to 15. Unfortunately, I have to decline this reviewer's suggestion because this so-called imbalance was deliberately sought by me. I have made a conscious and deliberate effort to present the fundamental chemical side of the problem, which can be successfully used by readers in elucidating the chemistry and mechanisms of lipid peroxidation in muscle foods. This section also aims to clarify the discussion in the following sections of the manuscript by providing explanations of a fundamental scientific nature. Anyway, thanks again to dear Reviewer 1 for the critical comments!
Line 493, Point 5.6, addressing Singlet Oxygen, requires an explanatory legend for the accompanying figure.
Thank you, it is corrected. Firstly, an explanatory legend for the accompanying figure on line 493 was added: “Photochemical pathway for singlet oxygen generation”. Following a similar recommendation by Reviewer 2, a correction was made in equation 19, since the product is singlet oxygen (1O2)! I decided to eliminate the equations for reactions 20 and 21. In their place, a short text has been added that describes the further course of chemical reactions to form hydroperoxide isomers. An appropriate new literature source is cited in the references. Respectively, reaction numbers 22 and 23 from the next page to section 5.7. Haemoproteins were adjusted on the 20-th and 2-st.
Line 583 suggests replacing "Introduction" with "Background" as a subheading in the main body of the paper for better alignment with scholarly conventions.
Thank you so much for this suggestion. It is now changed.
Line 867 acknowledges rosemary's antioxidant potential but suggests expanding the discussion to include other proven plant extracts like for example grape pomace, basil, sage, oregano, thyme, and artemisia.
Thank you so much for this question. The additional text was added and many new references are used.
Line 939 recommends placing references appropriately rather than combining them into one reference.
Thank you very much for this offer. I accept it. In this regard, some significant changes in the cited references are this paragraph as follows:
Eleventh references from old numbers 271 to 281 were removed:
271.Losada-Barreiro, S.; Bravo-Díaz, C. Free radicals and polyphenols: The redox chemistry of neurodegenerative diseases. Eur. J. Med. Chem., 2017, 133, 379-402. https://doi.org/10.1016/j.ejmech.2017.03.061 [New 340]
272.Coronado, S.A.; Trout, G.R.; Dunshea, F.R.; Shah, N.P. Antioxidant effect of rosemary extract and whey powder on the oxidative stability of wiener sausages during 10 months frozen storage. Meat Sci., 2002, 62, 217-224. https://doi.org/10.1016/S0309-1740(01)00249-2 [New 341]
273.der Veken, D.V.; Poortmans, M.; Dewulf, L.; Fraeye, I.; Michiels, C.; Leroy, F. Challenge tests reveal limited outgrowth of proteolytic Clostridium botulinum during the production of nitrate- and nitrite-free fermented sausages. Meat Sci., 2023, 200, 109158. https://doi.org/10.1016/j.meatsci.2023.109158 [was deleted]
- Bonifacie, A.; Promeyrat, A.; Nassy, G.; Gatellier, Ph.; Santé-Lhoutellier, V.; Théron, L. Chemical reactivity of nitrite and ascorbate in a cured and cooked meat model implication in nitrosation, nitrosylation and oxidation. Food Chem., 2021, 348, 129073. https://doi.org/10.1016/j.foodchem.2021.129073[New 342]
- [New 27@].Verde, C.; Giordano, D.; Bruno, S. NO and heme proteins: cross-talk between heme and cysteine residues. Antioxidants, 2023, 12, 321. https://doi.org/10.3390/antiox12020321[New 343]
- .[New 27@].Habib, S.; Ali, A. Biochemistry of Nitric Oxide. Ind. J. Clin. Biochem., 2011, 26, 3-17. https://doi.org/10.1007/s12291-011-0108-4[New 344]
- Kopf, S.H.; Henny, C.; Newman, D.K. Ligand-enhanced abiotic iron oxidation and the effects of chemical versus biological iron cycling in anoxic environments. Environ. Sci. Technol., 2013, 47, 2602-2611. https://doi.org/10.1021/es3049459[New 345]
- Sarraga, C.; Carrerras, I.; Regueiro, J.A.G. Influence of meat quality and NaCl percentage on glutathione peroxidase activity and values for acid-reactive substances of raw and dry-cured Longissimus dorsi. Meat Sci., 2002, 62, 503-507. https://doi.org/10.1016/S0309-1740(02)00039-6[New 346]
- Lehnert, N.; Kim, E.; Dong, H.T.; Harland, J.B.; Hunt, A.P.; Manickas, E.C.; Oakley, K.M.; Pham, J.; Reed, G.C.; Alfaro, V.S. The biologically relevant coordination chemistry of iron and nitric oxide: electronic structure and reactivity. Chem. Rev., 2021, 121, 14682-14905. https://doi.org/10.1021/acs.chemrev.1c00253[New 347]
- Smith, C.; Marletta, M.A. Mechanisms of S-nitrosothiol formation and selectivity in nitric oxide signalling. Cur. Opin. Chem. Biol., 2012, 16, 498-506. https://doi.org/10.1016/j.cbpa.2012.10.016[New 348]
- Huang, X.; Groves, T. Oxygen activation and radical transformations in heme proteins and metalloporphyrins. Chem. Rev., 2018, 118, 2491-2553. https://doi.org/10.1021/acs.chemrev.7b00373[New 349]
The reference under old number 273 was not replaced with a new one!
273.der Veken, D.V.; Poortmans, M.; Dewulf, L.; Fraeye, I.; Michiels, C.; Leroy, F. Challenge tests reveal limited outgrowth of proteolytic Clostridium botulinum during the production of nitrate- and nitrite-free fermented sausages. Meat Sci., 2023, 200, 109158. https://doi.org/10.1016/j.meatsci.2023.109158
Ten new literary sources have been used in their place. New numbers indicated those references after the sentences in which they quote.
NEW-340. Mynul Hasan Shakil, Anuva Talukder Trisha, Mizanur Rahman, Suvro Talukdar, Rovina Kobun, Nurul Huda, Wahidu Zzaman, Nitrites in Cured Meats, Health Risk Issues, Alternatives to Nitrites: A Review. Foods 2022, 11(21), 3355; https://doi.org/10.3390/foods11213355
NEW-341. Bruna Fernandes Andrade, Angélica Souza Guimarães, Lorrany Ramos do Carmo, Marcelo Stefanini Tanaka, Paulo Rogério Fontes, Alcinéia de Lemos Souza Ramos, Eduardo Mendes Ramos, S-nitrosothiols as nitrite alternatives: Effects on residual nitrite, lipid oxidation, volatile profile, and cured color of restructured cooked ham. Meat Science, Volume 209, March 2024, 109397. https://doi.org/10.1016/j.meatsci.2023.109397
NEW-342. Marco d’Ischia, Alessandra Napolitano, Paola Manini, Lucia Panzella, Secondary targets of nitrite-derived reactive nitrogen species: nitrosation/nitration pathways, antioxidant defence mechanisms and toxicological implications. Chem. Res. Toxicol. 2011, 24, 12, 2071–2092. https://doi.org/10.1021/tx2003118
NEW-343. J. A. Perez-Alvarez and J. Fernandez-Lopez, Chapter 4. Chemical and biochemical aspects of color in muscle foods. In: Handbook of Meat, Poultry and Seafood Quality, Edited by Leo M. L. Nollet, Blackwell Publishing, Blackwell Publishing Professional, Ames, Iowa 50014, USA, 2007, pp. 25-44.
NEW-344. Joseph Kanner, Stela Harel, Rina Granit, Nitric oxide, an inhibitor of lipid oxidation by lipoxygenase, cyclooxygenase and hemoglobin. Lipids, 1992, olume27, Issue1, 46-49. https://doi.org/10.1007/BF02537058NEW-345. 277. Irene Wood, Andrés Trostchansky, Homero Rubbo, Structural considerations on lipoxygenase function, inhibition and crosstalk with nitric oxide pathways. Biochimie, Volume 178, November 2020, Pages 170-180. https://doi.org/10.1016/j.biochi.2020.09.021
NEW-346. Jens K. S. Møller, Leif H. Skibsted, Nitric Oxide and Myoglobins. Chem. Rev. 2002, 102, 4, 1167–1178. https://doi.org/10.1021/cr000078y
NEW-347. Yafei Zhang, Xiaojing Tian, Yuzhen Jiao, Qiubo Liu, Ruonan Li, Wenhang Wang, An out of box thinking: the changes of iron-porphyrin during meat processing and gastrointestinal tract and some methods for reducing its potential health hazard. Critical Reviews in Food Science and Nutrition, 2023, 63:10, 1390-1405, https://doi.org/10.1080/10408398.2021.1963946
NEW-348. Huang, X., Ahn, D.U. Lipid oxidation and its implications to meat quality and human health. Food Sci. Biotechnol., 2019, 28, 1275-1285. https://doi.org/10.1007/s10068-019-00631-7
NEW-349. Małgorzata Karwowska, Anna Kononiuk, Karolina M. Wójciak, Impact of sodium nitrite reduction on lipid oxidation and antioxidant properties of cooked meat products. Antioxidants 2020, 9(1), 9. https://doi.org/10.3390/antiox9010009
Line 941. Why only the technological methods are discussed? It will be interesting to mention briefly other methods used for inhibition of lipid peroxidation in meat muscles. I think that some of these details can be found here (https://doi.org/10.3390/foods12214001), based on the findings from the literature.
Other methods besides the technological ones could be:
Enzyme Inhibitors involved in lipid peroxidation, such as lipoxygenase inhibitors. These can help disrupt the enzymatic initiation of peroxidation.
Irradiation reduces microbial load, extending the shelf life of muscle foods and indirectly inhibiting lipid peroxidation.
Heat treatment during processing to inactivate enzymes and reduce the likelihood of lipid peroxidation.
Microbial fermentation can produce bioactive compounds and metabolites that act as antioxidants and inhibit lipid peroxidation.
Thank you very much for these suggestions. It is possible to discuss methods other than technological ones. My original idea in determining the concept and structure of this manuscript was to draw the attention of food technologists to good production and practices in meat processing. Very deliberate section 6.4.3. is entitled Technological Methods for Inhibition of Lipid Peroxidation in Muscle Foods.
Thanks again for the proposal, but if you allow me, we will leave this extension of the manuscript for a future book or monograph.
Another observation is that this part is mostly focused on fish meat, and the rest of it (chicken, pork, and beef is neglected). Addressing the neglect of chicken, pork, and beef in this section will enhance it.
Thank you to Reviewer 1 for the critical notes and suggestions for improving this part of the manuscript. With all my respect for the findings of the respected Reviewer 1, but I do not consider it justified criticism that the section is most focused on fish meat, and chicken, pork and beef are neglected. In support of this statement, I enclose this part of the manuscript, in which with a yellow font I pointed to the discussion about poultry (white) meat and its products, with red font - red meat and sausages, incl. Beef, pork and more, and with a heavenly son - fish and fish products.
I would like to point out that glazing is a specific method of preventing oxidation processes and microbial cross-contamination specific and applied only to frozen fish! Insofar as the fresh fish, which contains significantly larger quantities of polyunsaturated lipids and it is more unstable to oxidative spoilage, including rancidity, in the literature discusses the possibilities of replacing glazing with new packaging techniques, edible films and coatings.This is the reason why I have presented in this part of the manuscript data about fish and fish products.
On the other hand, vacuum impregnation of muscle foods is a relatively new method of canning. Fish and fishery products discuss most of the references offering such treatment.
Therefore, I will be adjusted in this part of the manuscript.For these reasons, I will be adjusted in this part of the manuscript.
The packaging technique plays a crucial role in maintaining the oxidative stability of mature salted anchovies [281] and beef [282] during cold storage. Poultry meat retains its inherent qualities when vacuum-packed [147]. Lipid peroxidation in baked broiler meat, particularly with an increased content of α-linolenic acid [157], and slides of chilled baked chicken breasts [152], is inhibited when subjected to vacuum packaging. MAP is recommended for fresh sea fish [283]. Variations in packaging materials significantly affect the limitation of lipid peroxidation in fish sausages. The sensory attributes of frozen and chilled meat are best preserved through vacuum packaging [153, 160]. When utilizing a modified atmosphere like carbon dioxide or nitrogen, lipid peroxidation progresses more slowly than in air storage, but TBARS levels are higher than in vacuum-packaged meat [151, 153]. The presence of carbon dioxide in the atmosphere restricts lipid peroxidation processes [140]. At -10℃, lipids exhibit no signs of rancidity for up to 12 months [158]. Smoked chicken legs packaged in a MAP can be stored for 25 days at 4°C [284]. MAP contributes to maintaining the colour, taste, and flavour while suppressing microbial growth during the storage of chilled beef cuts [153, 285], beef [153], ground beef [155], rabbit meat [286], roasted pork, and turkey meat [287]. To enhance the effects of packaging in a modified atmosphere and the inhibition of lipid peroxidation, Balev et al. [286, 287] suggest treating chilled beef with natural antioxidants. To boost oxidation stability, MAP pork grill sausages with the addition of rosemary, ascorbic acid, sodium lactate, and red sugar beet roots were combined [289].
Edible films and coatings. Over the past decade, researchers have dedicated their attention to exploring environmentally friendly solutions for packaging meat and fish. Proposals for innovative strategies involve the application of environmentally friendly, biodegradable, and edible biopolymer films and coatings to meat, poultry, and fish products. The primary objective is to restrict oxygen access and mitigate microbial cross-contamination in muscle foods [290]. Polysaccharides such as cellulose, carrageenan, chitosan, pectin, starches, gums, alginates, as well as proteins like milk, collagen, and soy, have been discussed for this purpose. Additionally, various lipids, including essential oils, waxes, plasticizers, emulsifiers, and resins, have been considered [29].
Another innovative approach involves enriching edible films and coatings with active components to extend shelf life, reduce moisture evaporation losses, limit pathogenic microorganism growth, and slow down the development of putrefactive microbial spoilage. Notably, this approach aims to inhibit oxidative processes in the lipids, proteins, and pigments of muscle foods [292]. Examples of composite films for food packaging include those containing grass carp collagen, chitosan, and lemon essential oil [293], collagen and carboxymethyl cellulose films enriched with Boxthom barberry (Berberis lyceum) root extract [294], and films made from collagen from tuna skin, chitosan, and ultrasound-modified polyphenols from pomegranate (Punica granatum L.) [295].
Shokraneh et al. [296] have identified the effects of collagen fibres and green tea extract on the quality of vacuum-packed sausages. Typically, such edible coatings and films are frequently discussed in the context of preserving fish freshness. Various fish species have been reported to benefit from different edible coatings and films. Examples include treating with a gelatine film with propolis extract [297], covering red sea bream (Pagrus major) with a composite coating of chitosan and collagen derived from the skin of the blue shark [298], treating fresh salmon fillet (Salmo salar) with a film containing salmon bone gelatine, chitosan, gallic acid, and clove oil [299], or applying an edible alginate coating and surface treatment with dry-distilled pink petal extract or L-ascorbic acid to paddlefish (Polyodon spathula Walbaum, 1792) [300].
Additional methods involve covering carp (Cyprinus carpio) fillets with antioxidants extracted from nutmeg, rosemary, thyme, ginger, marjoram, parsley, turmeric, basil, and ginger using water and ethanol [301]. There is also the application of biodegradable antibacterial, antioxidant, and pH-sensitive hydrogel films containing carboxymethyl cellulose, collagen from dried fish bladder, eucalyptus extract, or quercetin [302], and the use of an edible gelatine coating and Portulaca oleracea extract on fish sausages [303].
Glazing of frozen fish. The glazing of frozen fish has been applied to restrict oxygen access, prevent microbial contamination, and minimize moisture evaporation from the surface of frozen fish. In recent years, there has been a proposal to enhance frozen fish by incorporating natural antioxidant extracts. The quality of frozen bonito (Sarda sarda) fillets can be effectively maintained by glazing with sage extract (Salvia officinalis) [304]. Frozen curimbata (Prochilodus lineatus) fillets are glazed with water containing turmeric extract [305]. Glazing frozen Atlantic horse mackerel (Trachurus trachurus) involves the use of water containing octopus’s products [306]. Squid (Pholidoteuthis massyae) is glazed with water containing preservatives [307]. Frozen Atlantic mackerel (Scomber scombrus) benefits from glazing with water containing quinoa [308]. Glazing frozen bigeye tuna (Thunnus obesus) entails the use of water containing rosemary acid, bamboo leaves, and sodium lactate [309]. Finally, frozen Nile tilapia (Oreochromis niloticus) is glazed with water containing two types of Kauaʻi pricklyash (Zanthoxylum kauaense) [310].
Vacuum impregnation of muscle foods. A novel approach to inhibit lipid peroxidation in muscle foods involves vacuum impregnation at various pressures [311]. Demir et al. [312] discovered that employing vacuum impregnation as a pre-treatment for traditionally marinated beef m. Longissimus dorsi with onion juice improved meat tenderness by 28.25%, with no alteration in colour brightness (L*). Simultaneously, tyramine concentration decreased, and TBARS levels were significantly lower than the limit value of 0.58 mg MDA.kg-1, as indicated by Martinez et al. [313], beyond which a rancid taste and smell are established.
Vacuum impregnation is a more effective method for salting Russian sturgeon fillets compared to salting under atmospheric pressure [314]. Fillets subjected to vacuum impregnation exhibited lower levels of TVB-N (total volatile basic nitrogen) and protein carbonyls, measuring 15.91 mg.100 g-1 and 311.38 nmol.L-1, respectively, after 5 hours of salting.
Zhao et al. [315] demonstrated the preservation of better quality and inhibition of lipid peroxidation in vacuum-impregnated grass carp (Ctenopharyngodon idella) fillets covered with an edible coating of chitosan infused with three types of water-soluble polyphenolic extracts - pomegranate bark, grape seeds, and green tea. Vacuum impregnation proved most effective in fillets covered with an edible film infused with green tea extract, packed in sterile polyethylene bags in an air atmosphere. The TBARS in this sample increased from 0.35 to approximately 1.70 mg MDA.kg-1 after 12 days of storage at 4°C.
Furthermore, Zhao et al. [316] investigated the potential inhibition of protein oxidation by combining an edible coating of fish gelatine and grape seed extract with vacuum impregnation of tilapia (Oreochromis niloticus) fillets stored for 12 days at 4°C. This combination was found to retard protein oxidation. The observed data are attributed to the formation of disulfide bonds, reducing total sulfhydryl groups and lowering Ca2+-ATP-ase activity.
Line 1247, titled "Effect of Oxidized Muscle Foods on Human Health," could benefit from a more focused exploration of the actual health effects, potentially incorporating examples from clinical studies.
Thank you very much for these suggestions.
Naturally, the potential involvement of examples of clinical studies in the article of actual effects on human health is an appropriate recommendation. Unfortunately, the volume of the paper exceeds 55 pages. Such clinical studies have been done and the volume of information is huge. If we do this, it would cover an additional 25 - 30 elderly. In this sense, we are already going to the volume of a monograph.
Therefore, at this stage I will kindly decline and will not take advantage of the proposal of Reviewer 1 and accept it as a recommendation for a future article or chapter of a book.
Line 1303 needs review to ensure it conveys the intended message clearly.
Thank you so much for this suggestion. The end of the sentence was deleted and I think this sentence is clear.
Line 1593's final sentence should be rephrased as a recommendation for further research.
Thank you so much for this note. The final sentence was rephrased as a recommendation for further research:
Ultimately, the scientific community's focus should be on identifying appropriate technological solutions to reduce the risk of initiating, developing, and spreading lipid peroxidation in meat and fish foods
“Further studies are needed to formulate appropriate technological solutions to reduce the risk of chemical hazards caused by the lipid peroxidation processes in muscle foods.”
Pros: The paper is compelling, of high quality, and warrants consideration for publication in the Foods Journal.
Thank you very much for the Reviewer 1 conclusion.
Cons: While the paper is well-researched, the extensive focus on chemistry and reactions, particularly those already studied, may require adjustment. There's also a noted discrepancy between the first part, covering Lipid Peroxidation, and the second part, addressing the Impact on Quality, Safety, and Human Health.
Thank you so much for these comments. With all my respect to the expert Reviewer 1, I allow myself not to accept his/her statement that there is a discrepancy between the first part, covering Lipid Peroxidation, and the second part, addressing the Impact on Quality, Safety, and Human Health.
The author is encouraged to consider addressing these concerns for an even more refined manuscript.
Good luck!
The author once again expresses his gratitude to Reviewer 1 for his thorough criticism of the manuscript. I appreciate the review as very helpful in improving the quality of the article. I hope that after the corrections and additions made, according to the recommendations of the two reviewers, the respected Reviewer 1 will be satisfied with the quality of the revised manuscript and recommend that it to be published in the Foods journal?
Comments on the Quality of English Language
English language fine. No issues detected
Thank you very much the Reviewer 1 for the Quality of English Language of this paper.

Reviewer 2 Report
Comments and Suggestions for Authors
The paper exhaustively describes the process of peroxidation in meat, including fish. The paper gives complete and important information about the factors affecting initiation, propagation, and termination from a chemical point of view but also considers the possible effects on human health because of the presence of oxidized products from meat. The paper is well written and I include some minor comments to consider.
Line 73: Please revise the sentence, since singlet oxygen usually is more reactive and favors the formation of the alkyl radical.
Line 106: what is the difference between oxy-myoglobin and oxymioglobin? Is the author referring to reduced myoglobin? Oxymioglobin is Fe2+ and is in a reduced state, not oxidized. Please carefully revise.
Line 232: Lipoxygenase is mainly found in vegetables. how the peroxidation occurs in meats?
Line 253-258: enzymatic initiation in fish is very different from meat? This paragraph is redundant, unless is an introduction for subsection 3.2.
Line 329: Figure 4, please check that MDA is placed under the correct formula.
Line 492: In reaction (19), singlet oxygen should not be the product of the reaction?
Line 597: worm over flavor? Is that correct?
Line 651-682: Why these paragraphs are focused only on poultry? In my opinion, it would be better a continued writing avoiding the subtitles in each paragraph. It applies to the whole 6.3. section.
Line 850: It would be great to include examples or substances considered as strong, weak, and synergistic. For example, the author indicates that polyphenols are strong antioxidants, what about amines? Are these substances also strong antioxidants? Or weak?
Line 1053: Please correct “Mayard” in the title by Maillard.
Line 1162: Maillard
In the health issues section: although the author considers the potential effect of oxidized products in several illnesses, it would be great if the author could include more information about how possible it is that radicals and oxidation products from meat can be absorbed by the human gastrointestinal tract and circulate along the body.
Comments on the Quality of English Language
The English grammar and style are ok, with some minor details.
Author Response
REVIEWER 2
Comments and Suggestions for Authors
The paper exhaustively describes the process of peroxidation in meat, including fish. The paper gives complete and important information about the factors affecting initiation, propagation, and termination from a chemical point of view but also considers the possible effects on human health because of the presence of oxidized products from meat. The paper is well written and I include some minor comments to consider.
Thank you very much for taking the time to review this manuscript. Thank you also for the kind words and positive evaluation of the manuscript, in general, as well as for the recommendations made for its improvement. Below I have provided my responses and opinions regarding your questions and comments.
Line 73: Please revise the sentence, since singlet oxygen usually is more reactive and favors the formation of the alkyl radical.
I thank Reviewer 2 for the precise reading of the text and for the remark made. What I wanted to say is exactly the opposite of what I wrote. When writing, I forgot to write the particle "not". I accept this remark. A new revision of the sentence has been made, with the added words indicated in blue font, and the word that is omitted in yellow font crossed out: “In a singlet basic condition, the formation of an alkyl radical (L) is not impeded hindered by this spin barrier [4].”
Line 106: what is the difference between oxy-myoglobin and oxymioglobin? Is the author referring to reduced myoglobin? Oxymioglobin is Fe2+ and is in a reduced state, not oxidized. Please carefully revise.
I thank Reviewer 2 for the precise reading of the text and for the remark made. Again, this is a technical inaccuracy, probably in the translation by a specialist in English philology who is not an expert in the field of the article. I accept the remark. This sentence comments on oxymyoglobin and oxyhaemoglobin. A new revision of the sentence has been made, as I have indicated the added word - "oxyhaemoglobin" in blue font, and the word that has been dropped – second "oxymyoglobin" in yellow font: “2) Through auto-oxidation of oxy-myoglobin and oxymyoglobin oxyhaemoglobin: (both …”
Line 232: Lipoxygenase is mainly found in vegetables. how the peroxidation occurs in meats?
Thanks to Reviewer 2 for the question. The answer to the question is complex and has been discussed by a number of authors, such as:
new-45. Min, B.R.; Nam, K.C.; Cordray, J.C.; Ahn, D.U. Factors Affecting Oxidative Stability of Pork, Beef, and Chicken Meat. In Iowa State University Animal Industry Report 2008, 1st ed.; Leaflet, A.S. Ed.; Iowa State University, Ames, Iowa, USA, 2008, pp. 1-4. https://doi.org/10.31274/ans_air-180814-1046
new -46. Wang, T.; Hammond, E.G. Chapter: 5 Lipoxygenase and Lipid Oxidation in Foods. In Oxidation in Foods and Beverages and Antioxidant Applications, 1st ed.; Decker, E.A. Ed.; Elsevier. Woodhead Publishing Limited, Sawston, Cambridge, UK, 2010, pp. 105-121. https://doi.org/10.1533/9780857090447.1.105
new -47. Grossman, S.; Bergman, M.; Sklan, D. Lipoxygenase in chicken muscle. J. Agric. Food Chem. 1988, 36, 1268-1270. https://doi.org/10.1021/jf00084a035
new -48. Min, B.; Nam, K.C.; Cordray, J.; Ahn, D.U. Endogenous factors affecting oxidative stability of beef loin, pork loin, and chicken breast and thigh meats. J. Food Sci., 2008, 73, C439-C446. https://doi.org/10.1111/j.1750-3841.2008.00805.x
new -49. Jin, G.; Zhang, J.; Yu, X.; Lei, Y.; Wang, J. Crude lipoxygenase from pig muscle: Partial characterization and interactions of temperature, NaCl and pH on its activity. Meat Sci., 2011, 87, 257-263. https://doi.org/10.1016/j.meatsci.2010.09.012
I think the reviewer's remark is related to the three literary sources cited in the manuscript - 12, 44 and 45. I fully accept the point and have re-edited parts of this paragraph citing new relevant references that discuss exactly how lipoxygenase peroxidation occurs in meat, but muscle tissue? Naturally, this changes the numbering of the cited references in the text, as you will notice with other corrections made to the manuscript.
Line 253-258: enzymatic initiation in fish is very different from meat? This paragraph is redundant, unless is an introduction for subsection 3.2.
After considering the remark of Reviewer 2, I accept the conclusion made. As a result, the text from lines 253-258 was deleted.
Line 329: Figure 4, please check that MDA is placed under the correct formula.
Thanks to Reviewer2 for the inaccuracy found. Indeed, the MDA text has moved forward a bit and does not sit exactly below the malondialdehyde formula. The diagram has been corrected!
Line 492: In reaction (19), singlet oxygen should not be the product of the reaction?
Thanks to Reviewer 2 for the identified non-conformity.
Indirectly, the transition of molecular oxygen from the triplet (3O2) to the singlet (1O2) state can be achieved using photosensitizers (light-absorbing molecules). The photosensitizer in its ground state 0S, absorbing light, passes into a singlet excited state 1S, which can pass without emission into a triplet state 3S (i.e., equation 18 in the paper which is true). If this 3S triplet state is energetically rich enough and the photosensitizer is close enough to the triplet oxygen (3O2) molecule, an energy transfer process can occur and singlet oxygen (1O2) can be generated. In doing so, the photosensitizer is regenerated to its original ground state.
Equation 19 was corrected because the product is singlet oxygen (1O2)!
Equations for reactions 20 and 21 were deleted. In their place, a short text was added. It describes the further course of the chemical reactions for the formation of hydroperoxide isomers. An appropriate new literature source in references was cited.
Respectively, reaction numbers 22 and 23 from the next page to section 5.7. Haemoproteins were adjusted on the 20-th and 2-st.
Line 597: worm over flavour? Is that correct?
Thanks to Reviewer 2 for spotting the blunder. A technical inaccuracy was admitted by the translator. Instead of “warmed-over flavour (WOF)” it was written: “worm over flavour (WOF)”. The term has been corrected!
Line 651-682: Why these paragraphs are focused only on poultry? In my opinion, it would be better a continued writing avoiding the subtitles in each paragraph. It applies to the whole 6.3. section.
Thanks to Reviewer 2 for the questions and recommendations made.
Paragraphs from lines 651-682 do not focus only on poultry. Perhaps the respected reviewer was left with a similar impression from the cited sources. We accept the remark, and in this part of the manuscript references are also cited that discuss red meat.
We also accept the reviewer's second suggestion to continue writing by avoiding subtitles in each paragraph. This discussion has been made not only in section 6.3, but also throughout the text where there are similar situations.
Line 850: It would be great to include examples or substances considered as strong, weak, and synergistic. For example, the author indicates that polyphenols are strong antioxidants, what about amines? Are these substances also strong antioxidants? Or weak?
Thanks to Reviewer 2 for the questions and recommendations made on line 850. Some example substances considered strong, weak and synergistic are included, such as polyphenols, amines, etc. Source 241 has been replaced with new references. My answer of the Reviewer’s question is follows:
The action of antioxidants depends on two factors: a) the activity of the free radical it forms (AO●), mainly dependent on the rate of its interaction with the substrate, and b) the rate of interaction between the antioxidant and peroxide radicals (RO and RO2). The first factor characterizes the strength of a particular antioxidant, while the second factor characterizes its effectiveness. The strength and effectiveness of an antioxidant are determined not only by its chemical properties but also by the properties of the oxidized substances.
The mechanism of action of different antioxidants depends on their chemical structure and nature. For example, phenols contain a mobile hydrogen atom. The molecule of such antioxidants can be represented by the expression InH, where H represents the mobile hydrogen atom. The oxidation of the phenol involved in inhibiting lipid peroxidation begins with the release of a hydrogen atom, forming a phenoxy radical.
The participation of strong antioxidants in side reactions is minimized. At the same time, the induction period of the oxidation reaction is directly proportional to the starting concentration of the antioxidant. At sufficiently high concentrations of the antioxidant, all peroxide radicals are replaced by its inactive radicals (AO), due to which the rate of oxidation is almost equal to zero.
With weaker antioxidants, the formed radicals (AO) also participate in other reactions, which is why the rate of oxidation is suppressed, but practically does not stop.
The effect of strong phenolic antioxidants is due to their free hydroxyl groups. When these groups are blocked, their antioxidant effect completely disappears. Electron-donor groups (methyl, methoxy, etc.) in o- and p-position significantly increase their antioxidant activity, and electron acceptor groups (nitroso-, carboxyl, etc.) decrease it. The effect of antioxidants increases as their concentration increases, but above a certain concentration, the so-called "inversion" of their action occurs. This effect is explained by an increase in the rate of decomposition of hydroperoxides into free radicals under the action of the antioxidant, such as e.g. the case of tocopherols. Therefore, the concentration of antioxidants usually does not exceed certain limits - in the case of tocopherols, 0.02 - 0.03%.
The most important representatives of strong antioxidants are phenolic compounds. Their action is explained by the interruption of the reaction chain as a result of their interaction with the active radicals R and RO leading the chain of oxidation, and their transformation into molecular products, in which, according to the principle of indestructibility of free valences, new inactive radicals of antioxidants are formed, which are not able to continue the chains.
Chain oxidation reactions can be inhibited not only by increasing the rate of chain termination, as is the case with the action of phenolics and some so-called "weak" antioxidants (quinones, amines), but also by reducing the rate of free radical formation through degenerate chain branching reactions. This can be done by introducing into the oxidizing environment substances capable of reacting with hydroperoxides without the formation of free radicals. Thus, hydroperoxides readily react with dialkyl sulfides. As a result, a sulfoxide is formed, capable of re-interacting with hydroperoxides. The product of this secondary reaction is a sulfone, which is formed at a much slower rate.
The mechanism of antioxidant action of this type of antioxidants is not chained, their effectiveness is significantly less than that of radical antioxidants. However, added in an oxidizing system together with phenolic antioxidants, they can significantly increase their effect. Some free acids and bases have a similar effect, causing the decomposition of hydroperoxides by an ionic mechanism without the formation of free radicals.
There are two opinions regarding the mechanism of action of amines as inhibitors of chain-radical reactions of lipid peroxidation. According to one of them, the inhibitory action of amines is due to the donation of a mobile hydrogen atom from their molecule to RO2● radicals, in which non-reactive compounds are formed. The least mobile atom in the amine molecule is the hydrogen from the amino group. According to the second opinion, in the process of inhibition, as an intermediate, a complex radical formed by the addition of RO2 is involved radical to the amine molecule, at the expense of a pair of free electrons from the nitrogen atom.
Quinones, capable of inhibiting lipid peroxidation processes, add free radicals to multiple bonds. When quinones are used to inhibit lipid end products, inactive or weakly active compounds are obtained.
When two inhibitors are used, a significant synergistic effect occurs when one of them interrupts the lipid peroxide chain reaction and the other destroys the peroxides. This phenomenon is explained by the fact that the two inhibitors, in addition to suppressing the oxidation of the main substance separately, mutually ensure each other from rapid depletion. The inhibitor that breaks the chain suppresses the formation of hydroperoxides, which prevents the rapid consumption of the other inhibitor - destroyer of peroxides. On the other hand, the second inhibitor destroys the peroxides, reduces the number of generating chains and thus preserves for a longer time the first inhibitor - breaking the chain.
When two antioxidants with different mechanisms of action are added together, a strong synergistic effect occurs.
It is possible to use as synergists compounds that do not have antioxidant properties or are very weak antioxidants. The addition of similar compounds significantly increases the effectiveness of the action of the other inhibitors. These include some polybasic organic hydroxycarboxylic acids, such as citric, tartaric, ascorbic, some amines, inorganic acids, e.g. phosphoric acid or its acid esters. Some amino acids, polyphosphates, cephalin, sulfhydryl and other compounds also appear as active synergists in the oxidation of fats in meat and fish foods.
The question of the mechanism of action of many of the synergists is complex. So e.g. some substances generally accepted as "true" synergists, by themselves do not have any antioxidant effect, but added to phenolic antioxidants, they enhance their effect. These include various organic and inorganic acids - phosphoric, citric, ascorbic, etc. The most plausible theory of their mechanism of action is Christiansen's theory, according to which they play the role of hydrogen donors to restore the spent antioxidant. In support of this theory is the fact that in the presence of an antioxidant, ascorbic acid is used up twice as fast, and the antioxidant itself - twice as slowly.
A synergistic effect in inhibited oxidation was also observed when two different antioxidants were used. This effect is particularly large when the two antioxidants are of different types - e.g. the first is a radical that breaks the oxidation chains, and the second is a peroxide decomposer.
A synergistic effect was also observed when using antioxidants of the same type, e.g. radical or peroxide scavengers.
A number of substances possess the property of binding heavy metals of variable valence in permanent chelate complexes and thus deactivating them as active pro-oxidants in relation to the oxidation process. In this way, they indirectly exhibit an antioxidant or synergistic effect in the process of fat oxidation. Obviously, deactivators show such an effect only in the presence of traces of heavy metals. Some authors separate the deactivators of metals into a separate class of antioxidants. Most often, active deactivators are a number of synergists - citric, tartaric, ascorbic acid, etc., which have a double action as synergists.
Synergists include a number of other polyhydroxy derivatives with appropriate placement of carboxyl and hydroxyl groups for the formation of chelate complexes, such as oxyacids, sorbitol, sucrose, etc. saccharides.
Some acids and bases also have a certain inhibitory effect on lipid peroxidation. They cause the decomposition of hydroperoxides by an ionic mechanism without forming free radicals. An example of such a compound is benzoic acid.
Line 1053: Please correct “Mayard” in the title by Maillard.
Thanks to Reviewer 2 for spotting the blunder. The name has been corrected!
Line 1162: Maillard
Thank you to Reviewer 2 for spotting the blunder. The name has been corrected!
In the health issues section: although the author considers the potential effect of oxidized products in several illnesses, it would be great if the author could include more information about how possible it is that radicals and oxidation products from meat can be absorbed by the human gastrointestinal tract and circulate along the body.
Thanks to the Reviewer 2 for the recommendation. The other respected Reviewer 1 made similar recommendation. I will answer the same way:
Naturally, the potential involvement of additional information on how it is possible to absorb the human gastrointestinal tract and circulate throughout the body is an appropriate recommendation. Unfortunately, the volume of the paper exceeds 55 pages. Such clinical studies have been done and the volume of information is huge. If we do this, it would cover an additional 25 - 30 elderly. In this sense, we are already going to the volume of a monograph.
Therefore, at this stage I will kindly decline and will not take advantage of the proposal of Reviewer 2 and accept it as a recommendation for a future article or chapter of a book.
Comments on the Quality of English Language
The English grammar and style are ok, with some minor details.
Thanks to Reviewer 2, for the assessment on the English grammar and style of the manuscript. After the revision, we hope the minor details commented on, have been removed.

Round 2
Reviewer 1 Report
Comments and Suggestions for Authors
The majority of the comments were addressed.